# Distinct functional determinants of influenza hemagglutinin-mediated membrane fusion

**Tijana Ivanovic[1,2,3]\*, Stephen C Harrison[2,4]**

[1]Department of Chemistry and Biochemistry, University of Colorado, Boulder, United States; [2]Department of Biological Chemistry and Molecular Pharmacology, Harvard Medical School, Boston, United States; [3]Department of Biochemistry, Brandeis University, Waltham, United States; [4]Howard Hughes Medical Institute, Harvard Medical School, Boston, United States

**Abstract** Membrane fusion is the critical step for infectious cell penetration by enveloped viruses. We have previously used single-virion measurements of fusion kinetics to study the molecular mechanism of influenza-virus envelope fusion. Published data on fusion inhibition by antibodies to the 'stem' of influenza virus hemagglutinin (HA) now allow us to incorporate into simulations the provision that some HAs are inactive. We find that more than half of the HAs are unproductive even for virions with no bound antibodies, but that the overall mechanism is extremely robust. Determining the fraction of competent HAs allows us to determine their rates of target-membrane engagement. Comparison of simulations with data from H3N2 and H1N1 viruses reveals three independent functional variables of HA-mediated membrane fusion closely linked to neutralization susceptibility. Evidence for compensatory changes in the evolved mechanism sets the stage for studies aiming to define the molecular constraints on HA evolvability.

**\*For correspondence:** ivanovic@brandeis.edu

## Introduction

Membrane fusion is the mechanism for directed interchange of contents among intracellular compartments. Carrier vesicles fuse with target organelles, secretory vesicles fuse with the plasma membrane, mitochondria fuse with each other. Enveloped viruses fuse with a cellular membrane to deposit their genomic contents into the cytosol.

Lipid bilayer fusion is a favorable process but with a high kinetic barrier (***Chernomordik and Kozlov, 2003***). Each of the examples of fusion just cited requires a protein catalyst. The SNARE complexes catalyze vesicle fusion (***Brunger, 2005***); mitofusins catalyze mitochondrial membrane fusion (***Chan 2012***); viral fusion proteins catalyze the fusion step essential for infectious cell entry (***White et al., 2008***, ***Harrison 2008***, ***2015***). The influenza hemagglutin (HA) is the best studied and most thoroughly characterized of the viral fusion proteins. Crystal structures determined in the 1980s and 1990s captured the fusion endpoints and showed that extensive structural rearrangements, triggered during entry by the low pH of an endosome, are part of the catalytic mechanism (***Wilson et al., 1981***, ***Skehel et al., 1982***, ***Bullough et al. 1994***, ***Chen et al.,1998***, ***1999***). Models for the fusion process then 'interpolated' intermediate states between these endpoints, supported by indirect evidence for specific features of these intermediates (***Figure 1***) (***Daniels et al., 1985***, ***Godley et al., 1992***, ***Carr and Kim, 1993***, ***Harrison 2008***, ***2015***).

Single-molecule techniques applied to studies of influenza virus fusion have yielded more direct information about the HA molecular transitions that facilitate it (***Floyd et al., 2008***, ***Imai et al., 2006***, ***Ivanovic et al., 2012***, ***Ivanovic et al., 2013***, ***Otterstrom and van Oijen, 2013***,

**eLife digest** Influenza (or flu) viruses can infect humans and other animals and can lead to life-threatening illness. To multiply, the virus particles must first enter a host cell. The final step in the entry process is the fusion of the membrane that surrounds the influenza virus with the membrane of the host cell. This event releases the core of the virus particle into the cell, where it can stimulate the cell to make more copies of the virus.

To ensure that membrane fusion takes place at the right place and time, influenza virus decorates the surface of its membrane with a protein called hemagglutinin. This protein senses cues provided by the target cell and then undergoes a series of transformations that lead to membrane fusion. During this process, hemagglutinin molecules insert into the target cell membrane to bring together the viral and cellular membranes.

In 2013, a group of researchers developed a computer simulation algorithm to study the events that lead to membrane fusion. In the model, the hemagglutinin molecules on a virus particle are activated at random to insert into the cell membrane. Now, Ivanovic and Harrison – two of the researchers from the earlier work – compared the predictions of this model to experimental data from previous studies of membrane fusion by influenza virus particles.

This approach shows that a substantial fraction of hemagglutinin molecules fail to contact the target-cell membrane and are permanently inactivated instead. Fusion nonetheless proceeds efficiently. Ivanovic and Harrison suggest that these inactive hemagglutinins provide an evolutionary backup store. For example, the proportion of hemagglutinins on a virus particle that insert into the cell membrane affects how fast fusion occurs and how sensitive the virus is to attack by host immune-system proteins called antibodies. Therefore, an ability to control how often hemagglutinins insert into the membrane could allow the virus to adapt to host immune responses. In the future, Ivanovic and Harrison's findings could aid the discovery of drugs that inhibit the entry of influenza into human cells.

*Otterstrom et al., 2014*, *Wessels et al., 2007*). The following picture emerged from experiments we described in 2013, in which we combined single-virion fusion observations with structure-guided mutation of HA (*Figure 1*) (*Ivanovic et al., 2013*). Trimeric HA 'spikes' densely cover the surface of an influenza virus particle. The contact zone between virus and target membrane (a supported lipid bilayer in the case of our experiments) contains between 50 and 150 HA trimers—a number that may be even larger for filamentous virions. When the pH drops below a critical threshold, individual HAs within the contact zone adopt an 'extended state', in which the fusion peptide at the N-terminus of $HA_2$ engages the target membrane, while the C-terminal transmembrane anchor remains embedded in the viral membrane. Note that the 'extended state' might represent an ensemble of folded-back conformations (*Figure 1A*). The probability of this stochastic event increases with proton concentration over the range at which groups on the protein titrate. A single HA trimer in the extended conformation cannot then fold back to its most stable, postfusion conformation, because of elastic resistance from the two membranes. Only when several neighboring HAs have extended and engaged can their joint action pull the two membranes together (*Figure 1B*). When the critical number of extended neighbors is present, foldback is cooperative and progression toward fusion is fast.

These observations led us to propose that the cooperativity of foldback comes simply from the mutual insertion of the cooperating HAs in both fusing membranes and that the number of HAs required is a function of the free energy released from individual HA fold-back events. When the total free energy is enough to overcome the 'hydration-force' barrier to merger (*Rand and Parsegian, 1984*), fusion can ensue. We called this a 'tug-of-war' mechanism—(N-1) trimers are not enough, but adding one more immediately precipitates a change, just as adding a critical extra team member will promptly snap a rope pulled against a fixed force. The team members need not touch each other as long as all are pulling on the same rope. An alternative model for cooperative action of fusion proteins comes from structural observations on alphavirus membrane fusion proteins, which suggest that a ring of five envelope-protein trimers might work as a single-unit fusion assembly

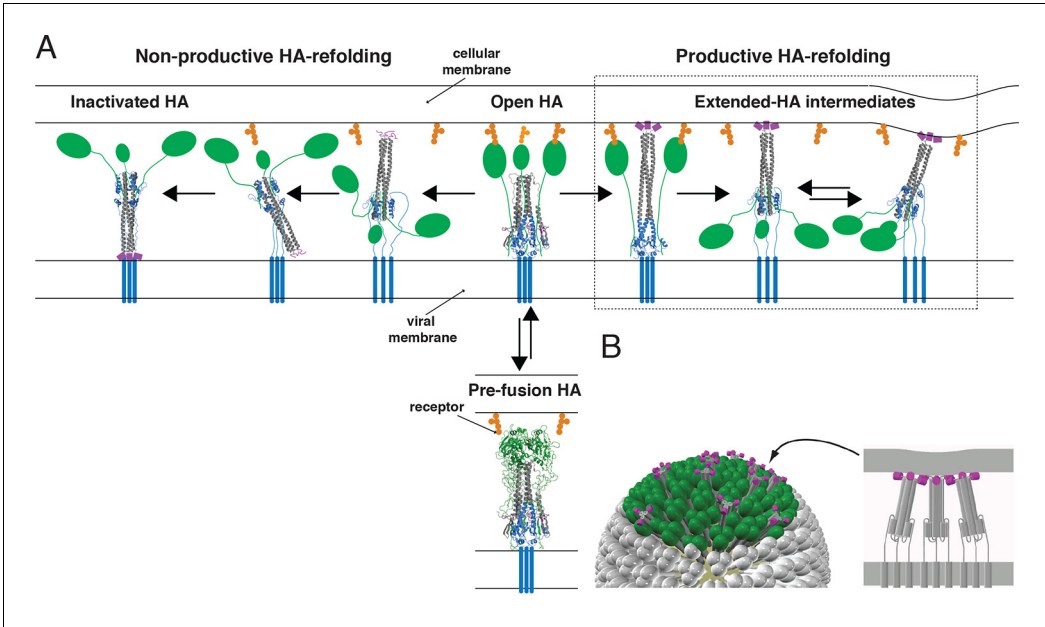

**Figure 1.** Productive and non-productive HA refolding, and membrane fusion by cooperative action of multiple, stochastically triggered HAs. (**A**) Proton binding increases the relative time HA spends in the 'open' conformation allowing fusion peptides to project toward the target membrane. $HA_1$ is shown in green and $HA_2$ in magenta (fusion peptides), gray (N-terminal 'half') and blue (C-terminal 'half'). *Right-hand arrow:* Productive HA refolding proceeds through an extended-intermediate state with fusion peptides inserted in the target membrane (*Ivanovic et al., 2013*). We illustrate a possibility that membrane-engaged HAs might represent an ensemble of folded-back conformations; the corresponding distance between the two membranes might fluctuate around a different value depending on how many HAs are cooperating. *Left-hand arrow:* Non-productive HA-refolding event occurs if HA assumes the low-pH form without target membrane engagement, resulting in loss of that HA as a potential fusion participant. (**B**) Individual-HA triggering and membrane insertion occur at random within a larger virion area that contacts the target membrane (~50 HAs shown in green are contained within this interface for a small, spherical influenza virion [*Ivanovic et al., 2013*]). Fusion ensues once a sufficient number of HAs – as needed to overcome the resistance of membranes to bending and apposition – are pulling jointly on the same membrane region (*Ivanovic et al., 2013*). 3D coordinates (PDB ID) used for displayed HA cartoons: the pre-fusion HA (2HMG), inactivated HA (1QU1); depicted intermediates are derived from a subset of either or both sets of coordinates (2HMG and/or 1QU1).

(*Gibbons et al., 2004*). This picture is a particular instance of mechanisms that require a defined, lateral interaction between participating proteins.

The probability of assembling a group of HA neighbors inserted into the target membrane depends on the fraction of *active* HAs. Some positions in the contact zone may be occupied by uncleaved $HA_0$, which cannot undergo the fusion-inducing conformational change (*Chen et al., 1998*), and others, by the viral neuraminidase, NA (although NA appears to cluster on one side of the budded particle: *Harris et al., 2006*, *Calder et al., 2010*, *Wasilewski et al., 2012*). Moreover, the fusion peptides of some HAs that do undergo the low-pH induced conformation change might fail to insert into the target membrane (*Figure 1A*). Exposure of unattached virions to low pH leads to inactivation, with the fusion peptides of rearranged HAs inserted back into the viral membrane, providing an experimental demonstration that non-productive conformational changes can indeed occur (*Weber et al., 1994*, *Wharton et al., 1995*). Simulations we used to derive kinetic parameters from single-virion fusion data can include estimates of inactive sites and unproductive events, and we show below the usefulness of this extension (*Figure 2*).

Addition of neutralizing antibodies can create additional inactive HAs. *Otterstrom et al. (2014)* recently used the single-virion assay together with fluorescently tagged IgGs or Fabs to study the occupancy required to achieve complete inhibition of viral fusion. They found that occupancies short of 100% were sufficient to reduce the yield of fusion to threshold. They concluded that these

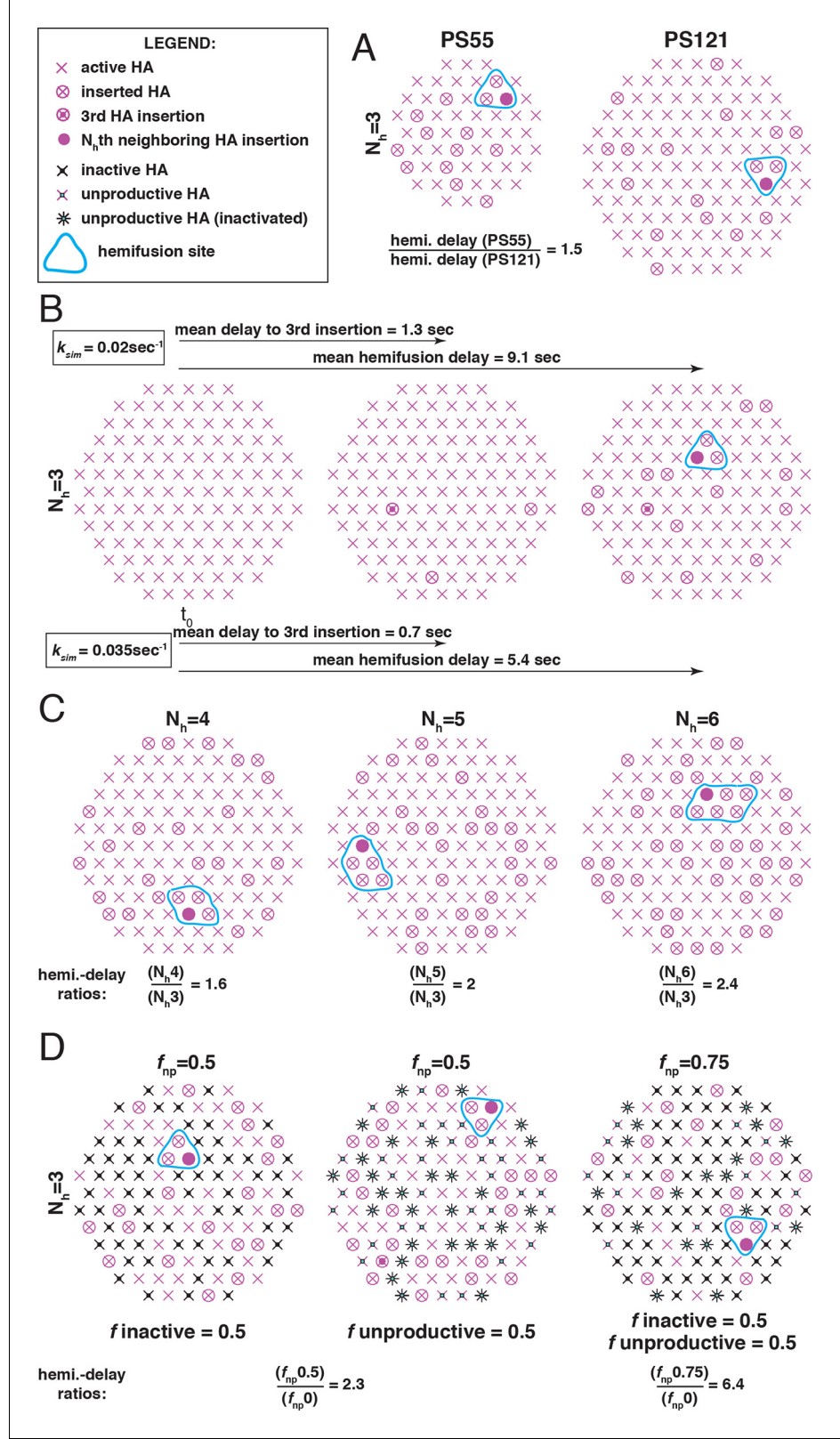

**Figure 2.** The functional variables of influenza membrane fusion modeled in this work. We modeled the kinetics and the extent of membrane fusion with the following parameters: (A) the number of HAs in contact with the target membrane (patch size, PS), (B) the rate ($k_{sim}$) of stochastic HA triggering, (C) the required number ($N_h$) of cooperating HA neighbors during fold-back (see *Figure 2—figure supplement 1* for the complete definition of

*Figure 2 continued on next page*

*Figure 2 continued*

six-mers ($N_h$ = 6) in the simulation), and (**D**) the frequency of inactive (*left*) or unproductive (*middle*) HAs, combined in the common parameter $f_{np}$ (*right*) as described in Materials and methods. Illustrations represent sample contact patches at the times of hemifusion except in panel B (*left and middle*), where they represent earlier time points. We compare the effects of various functional variables by either showing the ratios of mean hemifusion delays ($k_{sim}$-independent values) (A, C and D), or by directly showing mean hemifusion delays for two $k_{sim}$ values, and PS = 121, $N_h$ = 3 and $f_{np}$ = 0 (B). Our fusion model predicts that smaller patch size, lower $k_{sim}$, higher $N_h$, or higher $f_{np}$, will each increase hemifusion delay, and, with the exception of $k_{sim}$, will also, under certain conditions, reduce the theoretical fusion yield (see *Figure 3*).

The following figure supplement is available for figure 2:

**Figure supplement 1.** Definition of six-mers ($N_h$ = 6) in the simulation.

observations were consistent with the model we had proposed (*Ivanovic et al, 2013*) and that bound antibodies need simply to disrupt the network of potential neighbors rather than saturate the viral surface.

In the work we report here, we have used computer simulations to extend the analysis of fusogenic molecular events at the virus-target membrane interface (*Figure 2*) and compared the results with published single-virion experiments, including the recent studies of *Otterstrom et al. (2014)*. The extension includes an explicit parameter for the fraction ($f_{np}$) of 'non-participating surface elements' (those HAs that fail to engage and stochastically inactivate, those that have bound antibodies, those that are $HA_0$, and those sites in the model that might be occupied by NA) (*Figure 2D*). This analysis yields new conclusions concerning the course of viral fusion. We identify three independent functional variables of HA-mediated membrane fusion and find that virions from H3 and H1 influenza subtypes differ in at least two and possibly all three respects, and offer evidence for compensatory features of the evolved mechanism. The results illustrate the relative degrees of freedom available to influenza virus as it evolves in response to external pressures, whether from inhibitors, host immunity, or adaptation to replication in a new host species.

## Results

A step between separated lipid bilayers and full membrane fusion is formation of a hemifused intermediate (probably a 'hemifusion stalk'), in which the apposed leaflets have merged but the contents of the fusing compartments remain distinct (*Chernomordik and Kozlov, 2003*). In influenza virus fusion, lipid exchange, monitored by diffusion of a membrane-embeded hydrophobic dye, always precedes content exchange, monitored by diffusion of an internal hydrophilic dye (*Floyd et al., 2008*). The measurements of *Ivanovic et al. (2013)* and *Otterstrom et al. (2014)* therefore take hemifusion as their endpoint, and we do so in simulations described here.

### Simulations of molecular events at the virus-target membrane interface

We simulated stochastic HA triggering within the 'contact patch' between virus particle and target membrane, for patch sizes (PS) of 121 and 55 HA trimers (*Figure 3* and *Figure 3—figure supplement 1*), using the algorithm previously described (*Ivanovic et al., 2013* and Materials and methods). We included a range for the fractions of non-participating sites ($f_{np}$ – $HA_0$, NA, nonproductively refolded $HA_1$:$HA_2$) (*Figure 3A*) and allowed simulations to proceed to completion, i.e. until all the virions with potential to hemifuse had done so, or, until all HAs in the contact patch had extended and become either target-membrane engaged or inactivated (the highest value of $f_{np}$ we included yielded ~2% hemifusion). We defined the time of hemifusion as the moment at which the $N_h$th HA trimer joins a preexisting cluster of ($N_h$-1) HAs and determined, as functions of $f_{np}$, both the yield of hemifusion (percent of virions that hemifused) (*Figure 3B*) and the distribution of times from pH drop to hemifusion (*Figure 3C–E*). We ran the simulations for values of $N_h$ between 3 and 6. We previously concluded that $N_h$ = 2 yields data that do not agree with experiment results for H3 influenza (X31 and Udorn) (*Ivanovic et al., 2013*), and we provide here additional results to justify exclusion of this value in further analysis (*Figure 3—figure supplement 2*).

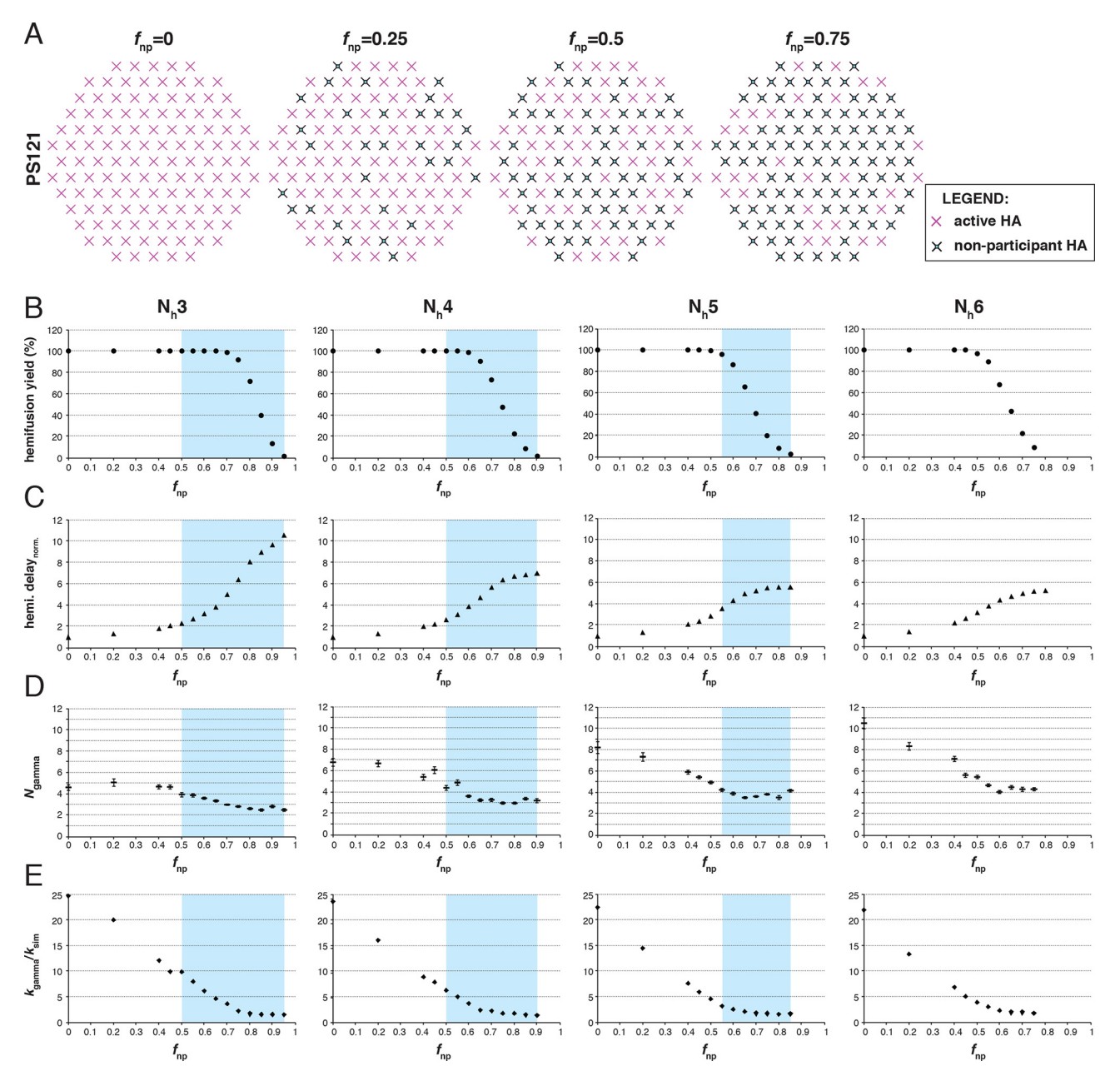

**Figure 3.** Effects of $f_{np}$ on hemifusion yield and kinetics for $N_h$ = 3–6 (PS = 121). (**A**) Illustration of simulated contact patches. (**B**) Hemifusion yield as a function of $f_{np}$. (**C**) Mean hemifusion-delay times normalized to $f_{np}$ = 0. (**D**) Parameter $N$ derived from fitting hemifusion delay distributions with the gamma probability distribution. Errors are 95% confidence intervals for the fit-derived values. (**E**) Parameter $k$ derived from fitting hemifusion delay distributions with the gamma probability distribution expressed as ratio with $k_{sim}$. By normalizing mean hemifusion-delay times and $k_{gamma}$, we obtained general trends, independent of the $k_{sim}$ value used in simulations. Plotted results are derived from simulations that yielded 1000–3000 hemifusion events. Blue shaded regions are estimates for the range of $f_{np}$ values consistent with $N_{gamma}$ values derived from experiment. The corresponding results for PS = 55 are shown in *Figure 3—figure supplement 1*. Refer to *Figure 3—figure supplement 2* for the simulation results for $N_h$ = 2 and both patch sizes. Refer to *Figure 3—figure supplement 3* for $N_{gamma}$ values derived from our previously published experimental datasets (*Ivanovic et al., 2013*).

The following figure supplements are available for figure 3:

**Figure supplement 1.** Effects of $f_{np}$ on hemifusion yield and kinetics for $N_h$ = 3–6 (PS = 55).

**Figure supplement 2.** Effects of $f_{np}$ on hemifusion yield and kinetics for $N_h$ = 2.

*Figure 3 continued on next page*

*Figure 3 continued*

**Figure supplement 3.** $N_{gamma}$ for pH-drop-to-hemifusion frequency distributions from previously published experiment data (*Ivanovic et al., 2013*).

The dependence of hemifusion yield and delay time on $f_{np}$ as $N_h$ varied over a reasonable range led us to conclude that the data in *Otterstrom et al. (2014)* could yield new information about these parameters (see what follows and the next results section, The gamma-distribution approximation). The simulations showed that the yield of hemifusion is relatively insensitive to the presence of inactive HAs for $N_h$ between 3 and 6 (*Figure 3B*). For $N_h = 3$, more than 70% ($f_{np} = 0.7$) of the sites on a virion surface must be unproductive or inactive in order to detect any reduction in fusion yield; for $N_h = 6$, we saw reduced yield whenever more than 50% of the sites lacked the potential to participate. The simulations also yielded relatively large increases in mean lag time to hemifusion for the tested range of $f_{np}$ values (*Figure 3C*). For $N_h = 3$, we found a tenfold, and for $N_h = 6$, a fivefold increase in mean time to hemifusion. In contrast to our simulation results, *Otterstrom et al. (2014)* observed sudden decreases in hemifusion yield for even the small numbers of bound antibodies or Fabs, and at most about a two-to-threefold increase in hemifusion lag times until complete inhibition of hemifusion. This difference could not be explained by a smaller patch size (*Figure 3—figure supplement 1*) and suggested to us that even for virions with no bound antibodies, a significant portion of surface sites lacked the potential to participate in fusion (i.e. the experiment was sampling from the right-hand portion of an entire theoretical inhibition curve). This qualitative conclusion is independent of the actual value of $N_h$ or $f_{np}$.

For all values of $N_h$, the mean hemifusion-lag times had the same overall dependence on $f_{np}$. As $f_{np}$ increased, a phase of relatively shallow dependence of the lag time gave way to a much stronger rate of increase, at about the same fraction at which the overall yield of hemifusion began to decline (compare *Figure 3B and C*). For $f_{np}$ values at which more than half of the simulated virions no longer yielded hemifusion, the lag time dependence reached a plateau. *Otterstrom et al. (2014)* indeed observed a plateau in mean hemifusion lag times as a function of increasing antibody or Fab concentration, thus offering experimental support for the prediction derived from the proposed mechanism of fusion (*Ivanovic et al., 2013*). Plateau occurs when additional reduction in the fraction of participating HAs is more likely to result in complete inhibition of hemifusion rather than further increase in the lag time. Indeed, for Fab concentrations in the plateau region for hemifusion delay, *Otterstrom et al. (2014)* found a continuing decrease in hemifusion yield as Fab concentrations increased. The result is intuitively reasonable. A high fraction of non-participating sites in a contact patch corresponds to a high probability that any particular HA will fail to engage the target membrane, either because it cannot change conformation (unprocessed $HA_0$ or inhibitor bound $HA_1:HA_2$) or because it has irreversibly inactivated (*Figure 3A*). When this probability becomes high enough, it becomes almost impossible to achieve $N_h$ membrane-engaged neighbors within a contact patch of fixed size (consider, for example, the number of ways one can fit $N_h = 6$ active HA neighbors within the contact patches illustrated in *Figure 3A* for different $f_{np}$ values).

## The gamma-distribution approximation

The gamma probability distribution represents the kinetics of a process in which N rate-limiting events of (uniform) rate constant $k$ occur in sequence. The first single-virion fusion experiments took $N$ from this representation as an estimate of the number of HAs required for hemifusion (*Floyd et al., 2008*). Subsequent comparison with simulation showed that the estimate is inaccurate when 100% of the virion surface can participate (*Ivanovic et al., 2013*). Dependence of $k$ on mutations that affect the docking of the fusion peptide in the pre-fusion trimer led to the conclusion that the rate-limiting step in the fusogenic conformational change is fusion-peptide exposure (*Ivanovic et al., 2013*).

To explore the effects of $f_{np}$ on the derived values of N and k, we fitted hemifusion-delay distributions from our simulations with gamma distributions (designating the parameters $N_{gamma}$ and $k_{gamma}$) (*Figure 3D and E*). We confirmed our previous conclusion that $N_{gamma}$ is an overestimate when all HAs in the contact patch are active (*Figure 3D*). We further found that simulation-derived $N_{gamma}$ approached the experimental values from previous studies of H3 viruses at high $f_{np}$ and $N_h = 3$–5. Except for a few specific data points, experimental values for $N_{gamma}$ are between 2 and 4

(see Materials and methods for summaries of previously published $N_{gamma}$ values and *Figure 3—figure supplement 3* for a subset of our own experimental data [*Ivanovic et al., 2013*]). Thus, considered in the context of our current simulations (*Figure 3D*), the relatively low experimental $N_{gamma}$ values support and generalize (beyond the experimental results of *Otterstrom et al. (2014)* the interpretation that even in the absence of targeted inhibition, a substantial portion of the sites on the virion surface lacks the potential to participate in fusion.

We further conclude that contrary to previous contentions (by us and others), $N_{gamma}$ alone does not distinguish among 3, 4 and 5 as the number of cooperating HA-neighbors because at high $f_{np}$, the theoretical $N_{gamma}$ values all closely match the experimental observations. On the other hand, the experimental values do rule out 6, for which the simulation derived $N_{gamma}$ was greater than 4, even for the highest $f_{np}$ values. Furthermore, in the simulations, $k_{gamma}$ derived from hemifusion-delay distributions was larger than the value for the rate constant ($k_{sim}$) corresponding to the probability used in the computation, but it approached this value at high $f_{np}$ (*Figure 3E*, plateau regions yield $k_{gamma}/k_{sim}$ between 1.5 and 2). In a large contact patch with a high fraction of participating HAs (low $f_{np}$ values), there are many ways to achieve $N_h$ neighbors (*Figure 3A*); as $f_{np}$ increases, that redundancy decreases, and $k_{gamma}$ becomes a better approximation to $k_{sim}$. $k_{gamma}$ does not reach the value of $k_{sim}$ even at the highest $f_{np}$ values, at which a majority of the virions that can hemifuse have only one way to reach hemifusion because they have only a single patch of $N_h$ active neighbors within a larger contact area containing mostly inactive or non-productively refolded HAs. Thus, to determine the rate constant for membrane engagement by individual HAs, one needs to determine the fraction of non-participating sites.

## Evidence for non-productive HA refolding

We have examined as follows the relative contributions to non-participating sites from NA, $HA_0$ and non-productive $HA_1$:$HA_2$ refolding. The clustered localization of NA on a virion and its surface occupancy of 10-15% (*Harris et al., 2006*, *Calder et al., 2010*, *Wasilewski et al., 2012*) lead us to expect NA to make only a very small contribution. In *Figure 4* and *Figure 4—figure supplement 1*, we show that the virions used in our previous experiments (*Ivanovic et al., 2013*) had fully processed HA and that the HAs had full potential to assume the low-pH induced conformation. We thus conclude that non-productive HA refolding is the major component of non-participating sites in our previous experiments. Given similar predictions for $f_{np}$ values based on experimental $N_{gamma}$ values from the preceding paragraph, this conclusion might well extend to other single-virion experiments of influenza membrane fusion (*Floyd et al., 2008*, *Otterstrom et al., 2014*), although we cannot formally conclude that here. For simplicity, however, in the subsequent set of analyses, we refer to non-participating sites in the absence of targeted HA inhibition as unproductive HAs, and their frequency on the virion surface as $f_{un}$.

## Fab inhibition of H3 HA

*Otterstrom et al. (2014)* studied inhibition of hemifusion by Fabs and IgGs of HA stem-directed antibodies. They determined that for H3N2 X31 virions, an average of 261 bound Fabs gave half-maximal hemifusion inhibition and that 493 Fabs inhibited hemifusion completely. (We consider only their Fab data here, to avoid potential complications from divalent binding of IgGs.) We simulated inhibition, taking 375 as the number of HAs per virion (1125 Fab sites) (see Materials and methods) (*Figure 5A*). We assumed random HA occupancy and postulated that a single bound Fab prevents the fusion transition of a trimer. We varied $f_{un}$ values and looked for fractions that gave 50% hemifusion-yield inhibition for 261 bound Fabs and near complete inhibition for 493 bound Fabs. For $N_h = 3$ and $N_h = 4$, we obtained essentially unique answers for $f_{un}$ (*Figure 5B and C*): 0.65 with $N_h = 3$, and 0.4 with $N_h = 4$. With $N_h = 5$, no condition was consistent with the measured values (*Figure 5D*). This treatment of the inhibition data has thus allowed us to determine possible pairs of values for the number of neighboring HAs required for hemifusion and the fraction of unproductive HAs. For somewhat reduced $f_{un}$ values, the data are also consistent with a smaller patch size (see *Figure 5—figure supplement 1*). This result makes intuitive sense because conceptually, a smaller patch size is like a larger patch size with more non-participating sites.

The following more complete analysis of the data in *Otterstrom et al. (2014)* favors the interpretation that for X31 H3 HA, three HA neighbors cooperate during fold-back. To facilitate comparison

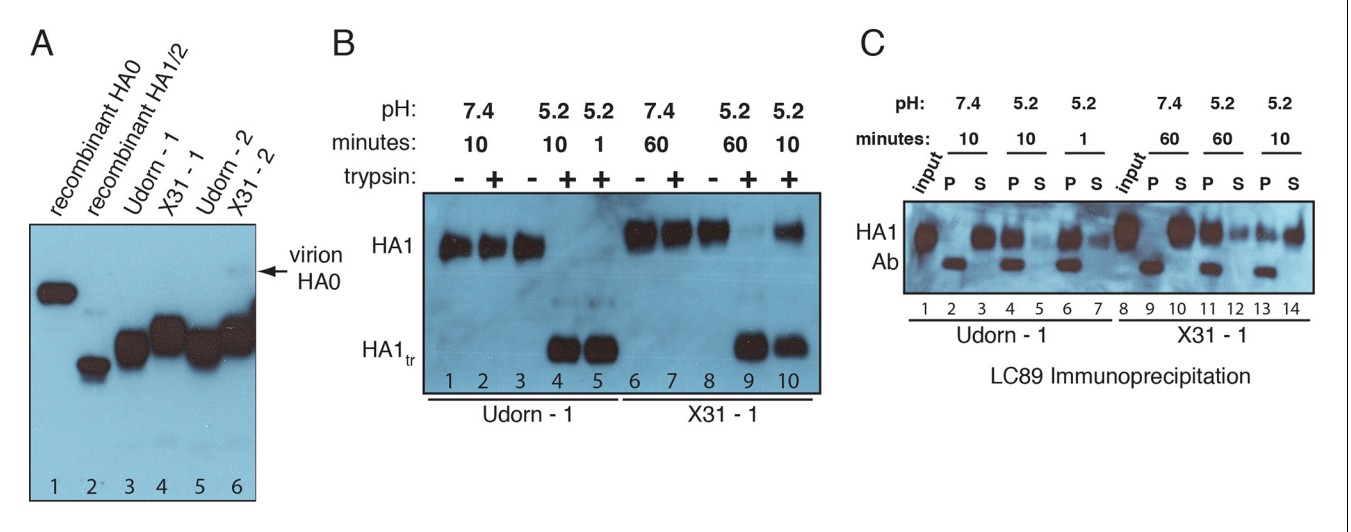

**Figure 4.** Complete processing of virion-associated HAs and complete conformational change at low pH. We show WT UdornHA-Udorn and X31HA-Udorn virions used in our previous single-virion fusion experiments (*Ivanovic et al., 2013*). SDS-PAGE and western blot of virions probed with $HA_1$-specific antibody that detects both $HA_0$ and $HA_1$ alone. (**A**) Recombinant X31 $HA_0$ and $HA_1$:$HA_2$ are included as a reference. The various HA forms appear to show varying levels of glycosylation resulting in different gel migration patterns. A trace amount of unprocessed $HA_0$ is apparent in only one of two X31HA-Udorn preparations (*lane 6*, band location marked with an *arrow*). (**B,C**) Virions were incubated in either neutral or pH5.2 buffer for indicated times at 37°C. (**B**) Virions were either loaded directly onto the gel or treated with trypsin prior to loading. Resistance to trypsin digestion of virion-HA incubated in neutral buffer is a control for pre-fusion HA integrity. $HA1_{tr}$ is the trypsin-resistant fragment of $HA_1$ (**C**) Virions were immunoprecipitated with LC89 antibody (specific for the low-pH form of $HA_2$ [*Wharton et al., 1995*]), and the entire bead-associated fraction (P) and the supernatant (S) were loaded onto separate lanes of the gel. Ab refers to the band corresponding to the heavy chain of the antibody used for immunoprecipitation, detected with the secondary antibody used in the western blot. Complete HA conversion to trypsin-sensitive form or to a form that can be immunoprecipitated with LC89 antibody is apparent by 1 min for Udorn HA and by 60 min for X31 HA. The conversion kinetics for X31-HA are disproportionately slower than its fusion kinetics (*Ivanovic et al., 2013*); see the Discussion for consideration of the consequences of these observations for the fusion mechanism. An analogous set of results for the second UdornHA-Udorn and X31HA-Udorn clones are shown in *Figure 4—figure supplement 1*.

The following figure supplement is available for figure 4:

**Figure supplement 1.** Complete processing of virion-associated HAs and complete conformational change at low pH.

with the reported data, we derived from simulations values for the yield of hemifusion, for the geometric mean of hemifusion-delay times, and for $N_{gamma}$ and $k_{gamma}$, as functions of the number of Fabs bound per virion (*Figure 6* and *Figure 6—figure supplement 1*). We carried out these simulations for the permitted $N_h$:$f_{un}$ pairs (obtained from the data in *Figure 5* and *Figure 5—figure supplement 1*) as we increased $f_{Fab}$ across the reported range. We adjusted $k_{sim}$ so that the geometric mean of the hemifusion delay times in the absence of any bound Fabs was ~30 sec, the value reported for H3N2 X31 virions under the conditions of the measurements in *Otterstrom et al. (2014)*. For either patch size, this procedure yielded values for $k_{sim}$ of 0.02 and 0.017 sec$^{-1}$ for $N_h = 3$ and $N_h = 4$, respectively (*Figure 6* and *Figure 6—figure supplement 1*).

*Figure 6C* shows that for $N_h = 3$, the mean hemifusion delay time in the simulation increased from ~30 to ~80 sec (a 2.7-fold increase) as the number of bound Fabs increased from zero to 500 (the latter corresponding to slightly under half occupancy). For $N_h = 4$, the delay time with 500 bound Fabs was 100 sec (a 3.6-fold increase). Again, the comparison is independent of patch size, as expected (see comment above) (*Figure 6—figure supplement 1*). *Otterstrom et al. (2014*; their Figure 3) reported a 2.6 ± 0.4-fold increase, i.e. a delay time of ~80 sec for 500 bound Fabs, in good agreement with the simulation for $N_h = 3$ (to facilitate comparison with our simulations, we plotted these published experimental data onto the panels in *Figure 6B–E*).

*Figure 6D* shows that for $N_h = 3$, $N_{gamma}$ was approximately equal to 3 and nearly independent of the number of bound Fabs, while for $N_h = 4$, $N_{gamma}$ fell from greater than 5, for no bound Fabs, to about 3 at higher Fab occupancies. *Otterstrom et al. (2014)* reported $N_{gamma}$ ~2.5, with little

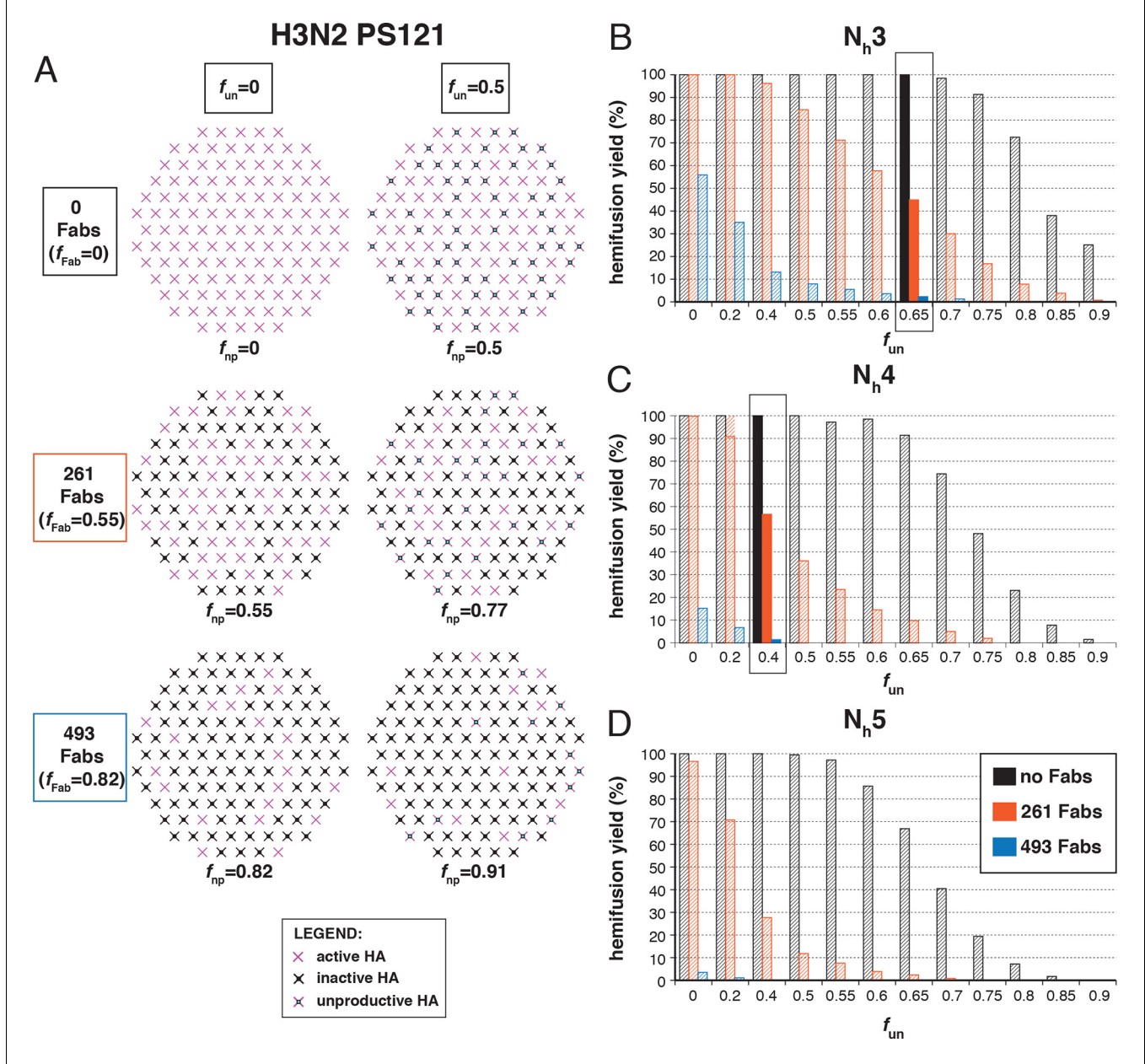

**Figure 5.** Hemifusion yield as a function of the fraction of unproductive HAs ($f_{un}$) for virions with no bound antibody and for those with 261 or 493 bound Fabs (PS = 55). (**A**) Illustrations of simulated contact patches. The frequency of Fab-bound HAs ($f_{Fab}$) and $f_{un}$ were combined in the parameter $f_{np}$ as described in Materials and methods. (**B–D**) The results for $N_h = 3$ (**B**), $N_h = 4$ (**C**), and $N_h = 5$ (**D**) were derived from simulations that yielded 1000-3000 hemifusion events. Non-zero $f_{un}$ values (boxed out regions in (**B**) and (**C**) are required to explain the experimentally observed number of Fabs required for half-maximal (261) and maximal (493) inhibition of H3N2 X31 influenza virus hemifusion (*Otterstrom et al., 2014*). Experimental data are inconsistent with $N_h = 5$. The corresponding results for PS = 55 are shown in *Figure 5—figure supplement 1*.

The following figure supplement is available for figure 5:

**Figure supplement 1.** Hemifusion yield as a function of the fraction of unproductive HAs ($f_{un}$) for virions with no bound antibody and for those with 261 or 493 bound Fabs (PS = 121).

dependence on Fab occupancy, again in better agreement with the $N_h = 3$ simulation results. We verified that the predicted 2-point drop in $N_{gamma}$ would be evident despite the uncertainty in fitting $N_{gamma}$ inherent in small datasets (*Figure 6—figure supplement 2*, H3N2 results). Furthermore, for

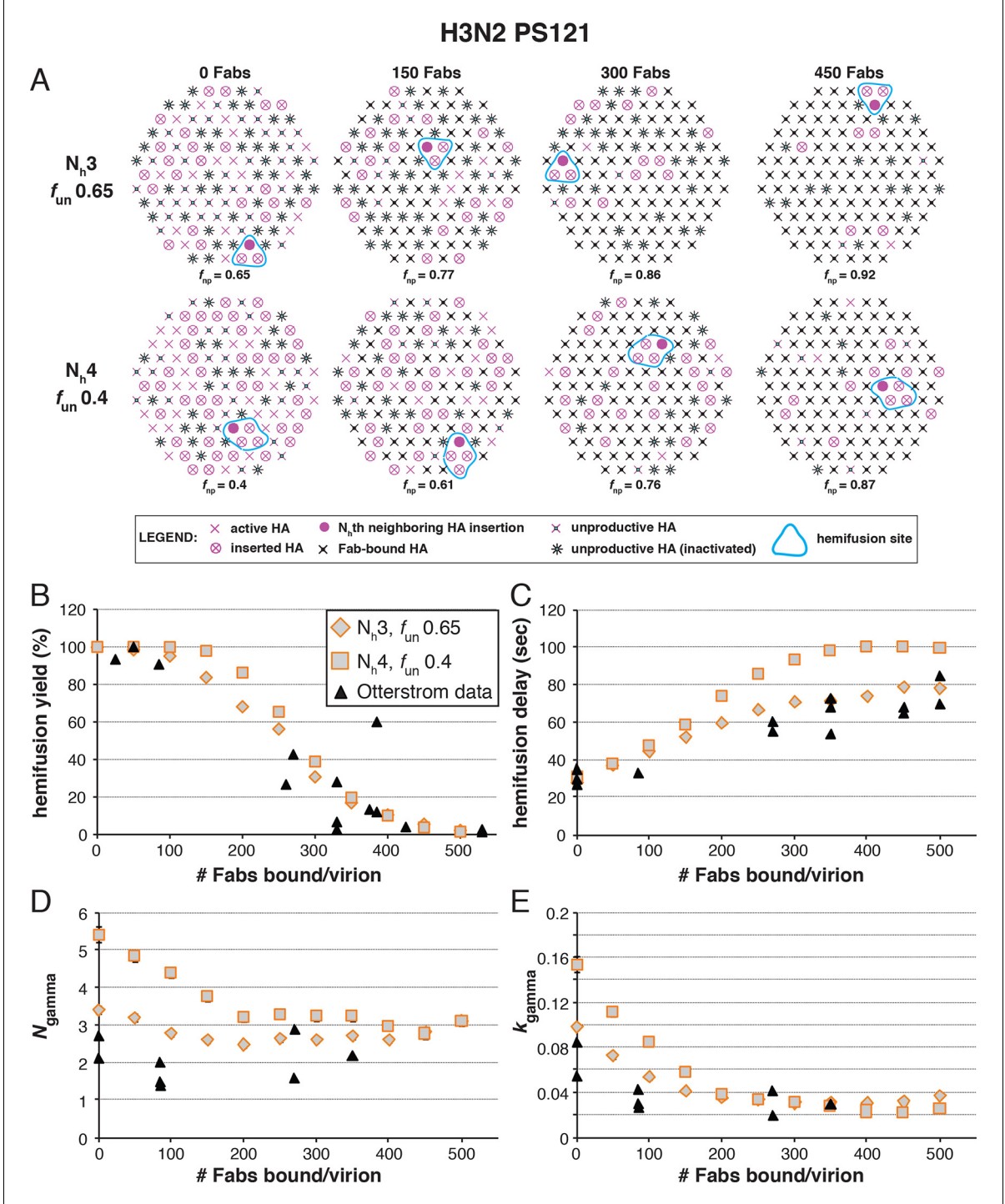

**Figure 6.** Effects of Fab binding on hemifusion yield and kinetics for given pairs of $N_h$ and $f_{un}$ (PS = 121). (**A**) Illustrations of simulated contact patches at the time of hemifusion for several $f_{np}$ values ($f_{un}$ was kept constant while $f_{Fab}$ was increased). (**B–E**) Comparison of simulation-derived results (1000–3000 hemifusion events) for hemifusion yield (**B**), hemifusion delay (geometric mean) (**C**), $N_{gamma}$ (**D**) and $k_{gamma}$ (**E**) with experimental data for H3N2 X31 influenza from *Otterstrom et al. (2014)* (*black triangles*). Experimental hemifusion yield data in (**B**) (their Figure 2C) were scaled so that the highest measured hemifusion yield value became 100% (i.e. each data point was multiplied by 4/3). The corresponding results for PS = 55 are shown in *Figure 6—figure supplement 1*. For simulations testing the effect of sample size on variability in $N_{gamma}$, see *Figure 6—figure supplement 2*. For a further test of the robustness of the conclusions derived from this figure, see *Figure 6—figure supplement 3*.

The following figure supplements are available for figure 6:

*Figure 6 continued on next page*

*Figure 6 continued*

**Figure supplement 1.** Effects of Fab binding on hemifusion yield and kinetics for given pairs of $N_h$ and $f_{un}$ (PS = 55).

**Figure supplement 2.** Effect of sample size on variability in $N_{gamma}$.

**Figure supplement 3.** Effects on our conclusions of potential error in the measurement of the number of Fabs needed for 50% hemifusion inhibition (#Fab$_{1/2hemi}$) for H3N2 X31 influenza virions.

$N_h$ = 3, $k_{gamma}$ from simulation showed a moderate (~threefold) drop from ~0.1 to ~0.03 sec$^{-1}$, again in much better agreement with the shown experiment values (*Otterstrom et al., 2014*) than the predicted ~fivefold drop in this value for $N_h$ = 4 (*Figure 6E*). We further tested the robustness of the above conclusions against potential uncertainty in the measured value for the number of Fabs (#Fab$_{1/2hemi}$) needed to achieve half-maximal hemifusion inhibition (*Figure 6—figure supplement 3*).

We conclude that $N_h$ = 3 gives very good agreement of simulation and experiment for several observed or derived parameters and a range of #Fab$_{1/2hemi}$ values. A consequence is that for H3N2 X31 virions under the experimental conditions of *Otterstrom et al. (2014)*, the rate constant ($k_e$) for the limiting kinetic step during productive HA extension corresponds to $k_{sim}$ for the combination of parameters that best fits all the observations (~0.02 sec$^{-1}$) (see above). Moreover, *Figure 6B* shows that to fit the observed data, all virions must have the potential to fuse (that is, the simulated yield of hemifusion in the absence of Fabs is 100%, when the simulations are run with the parameters that best fit all the observations). The yield of hemifusion for H3N2 X31 virions reported by *Otterstrom et al. (2014)* was about 60%, which thus calibrates the efficiency of the assay and the method of virion detection. The yield in our own earlier work on H3N2 X31 and Udorn particles was about 80% (*Ivanovic et al, 2013*).

## Fab inhibition of H1 HA

The number of bound Fabs required to inhibit fusion of H1N1 PR8 influenza virions in the experiments of *Otterstrom et al. (2014)* was substantially lower than for H3N2 X31 — on average, 74 Fabs for half-maximal inhibition and 248 Fabs for complete inhibition. This difference suggests either that PR8 viruses require more HAs for hemifusion or that non-productive conformational changes are more likely (or both). (Virion size was the same for the H3 and H1 strains, so patch-size difference is not the reason for their differential neutralization susceptibility.) Following the same procedure as above for H3 HA (*Figure 5*), we could find, for each value of $N_h$ between 3 and 6, a single value for $f_{un}$ that gave both 50% fusion-yield inhibition for 74 bound Fabs and near-complete inhibition for 248 bound Fabs (*Figure 7*). As expected, for somewhat reduced $f_{un}$ values, the data are also consistent with a smaller patch size (see *Figure 7—figure supplement 1*).

We proceeded to distinguish among the potential pairs of values for $N_h$ and $f_{un}$ as we did with the H3N2 X31 data (*Figure 8* and *Figure 8—figure supplement 1*). We carried out the simulations for each of the permitted $N_h$:$f_{un}$ pairs (obtained from the data in *Figure 7* and *Figure 7—figure supplement 1*), and calculated the various experimentally observed parameters as we increased $f_{Fab}$ until near complete hemifusion inhibition (*Figure 8A*). We adjusted the values for $k_{sim}$ so that the mean hemifusion delay time in the absence of bound Fab was about 46 sec, as determined by *Otterstrom et al. (2014)*. For either patch size, the corresponding $k_{sim}$ ranged from 0.029–0.037 sec$^{-1}$ for $N_h$ from 3–6. The simulated yield of hemifusion for no bound Fab varied from about 65–70% for $N_h$ = 5 or 6 to less than 50% for $N_h$ = 3 or 4 (panel B in *Figure 8* and *Figure 8—figure supplement 1*). *Otterstrom et al. (2014)* reported a 45% yield for H1N1; if we calibrate based on their yield for H3N2 of 60%, for which simulation indicates 100% (see above), we get a 'corrected' yield of 75%. Although approximate, this rescaling takes into account the experimental uncertainties that will make the observed yield lower than modeled by the simulation; for example, the program used to select virus particles will with some frequency pick non-particles (fluorescent spots) that will certainly fail to fuse (at least 7-9% in our published experiments: *Ivanovic et al, 2013*). Imperfections in the planar bilayer would prevent detection of potential fusion events from particles that might land on them (e.g., stick to glass exposed at a hole in the bilayer). Moreover, within the assumptions of

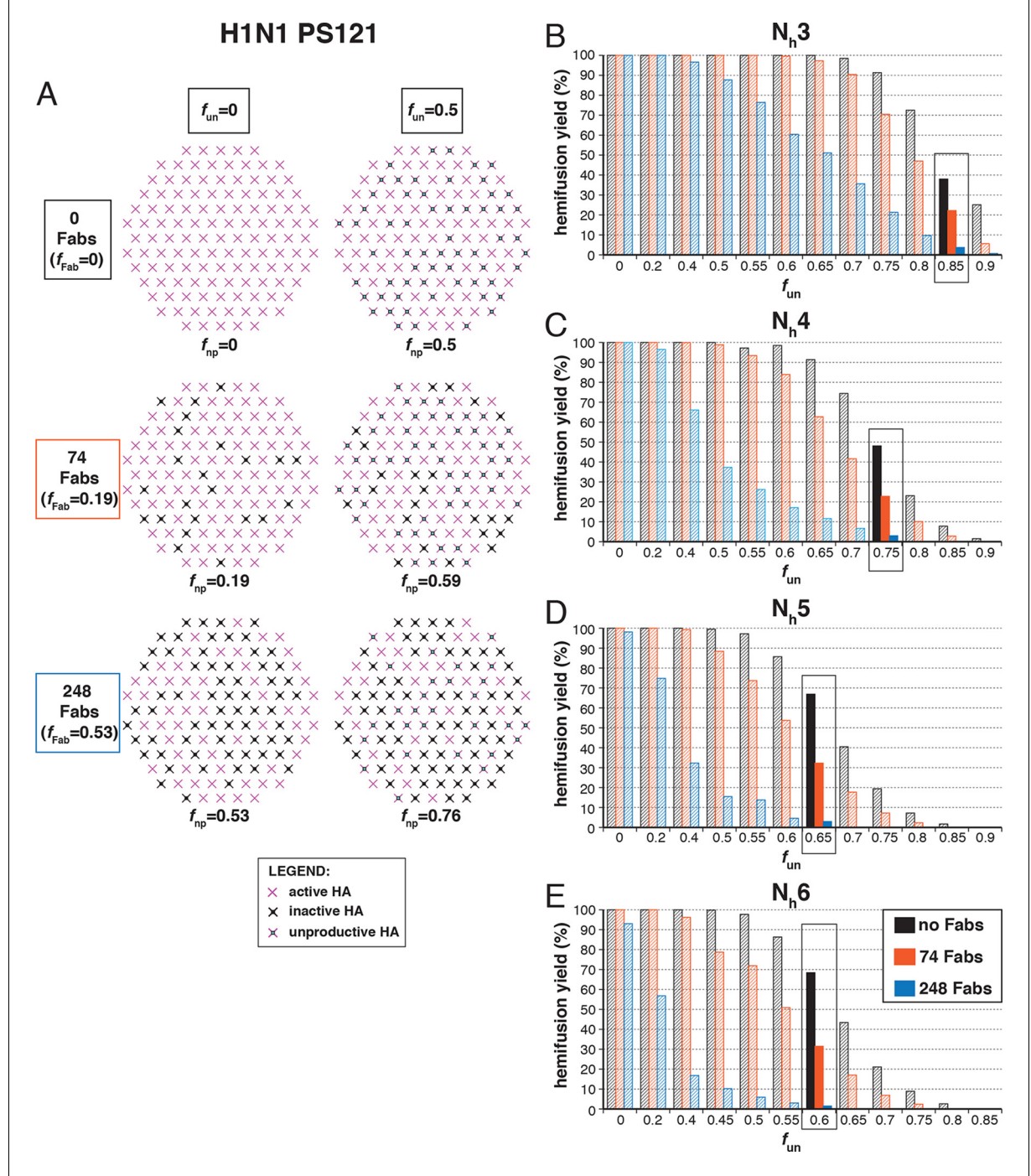

**Figure 7.** Hemifusion yield as a function of $f_{un}$ for virions with no bound antibody or those with 74 or 248 bound Fabs (PS = 121). (**A**) Illustrations of simulated contact patches. (**B–E**) The results for $N_h$ = 3 (**B**), $N_h$ = 4 (**C**), $N_h$ = 5 (**D**), and $N_h$ = 6 (**E**) were derived from simulations that yielded 1000 to 3000 hemifusion events. Non-zero $f_{un}$ values (boxed-out regions) are required to explain the experimentally derived number of Fabs required for half-maximal (74) and maximal (248) inhibition of H1N1 PR8 influenza virus hemifusion (*Otterstrom et al., 2014*). For different $f_{un}$ values, data are consistent with $N_h$ = 3–6. The corresponding results for PS = 55 are shown in *Figure 7—figure supplement 1*.

The following figure supplement is available for figure 7:

**Figure supplement 1.** Hemifusion yield as a function of $f_{un}$ for virions with no bound antibody or those with 74 or 248 bound Fabs (PS = 55).

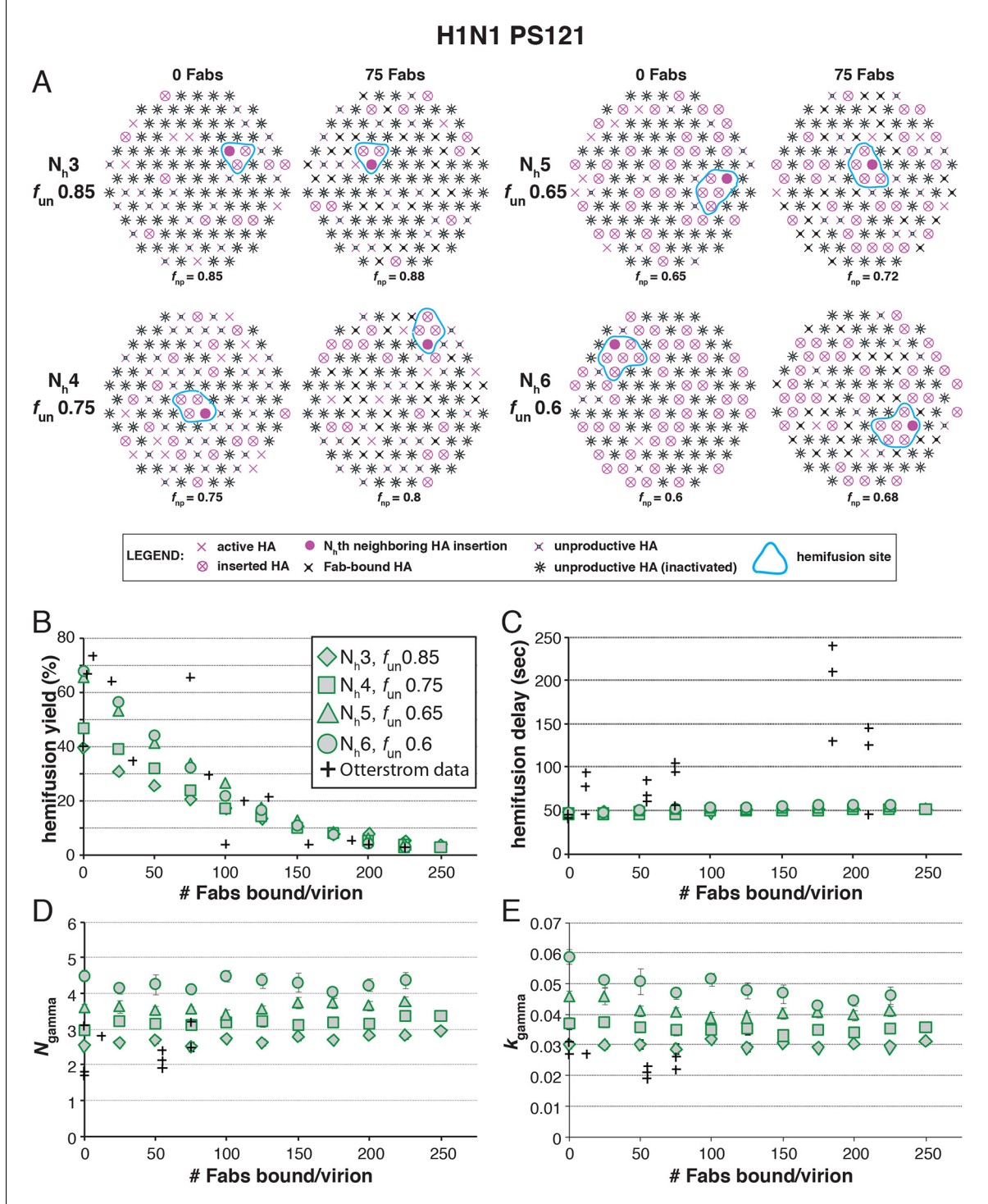

**Figure 8.** Effects of Fab binding on hemifusion yield and kinetics for given pairs of $N_h$ and $f_{un}$ (PS = 121) (**A**) Illustrations of simulated contact patches at the time of hemifusion for several $f_{np}$ values ($f_{un}$ was kept constant while $f_{Fab}$ was increased). (**B–E**) Comparison of simulation-derived results (1000–3000 hemifusion events) for hemifusion yield (**B**), hemifusion delay (geometric mean) (**C**), $N_{gamma}$ (**D**) and $k_{gamma}$ (**E**) with experimental data for H1N1 PR8 influenza from *Otterstrom et al. (2014)* (*black pluses*). Experimental hemifusion yield data in (**B**) (their Figure 2C) were scaled using the same factor as in *Figure 6B* (each data point was multiplied by 4/3). The corresponding results for PS = 55 are shown in *Figure 8—figure supplement 1*. For a further test of the robustness of the conclusions derived from this figure, see *Figure 8—figure supplement 2*.

The following figure supplements are available for figure 8:

*Figure 8 continued on next page*

*Figure 8 continued*

**Figure supplement 1.** Effects of Fab binding on hemifusion yield and kinetics for given pairs of $N_h$ and $f_{un}$ (PS = 55).
**Figure supplement 2.** Effects on our conclusions of potential error in the measurement of the number of Fabs needed for 50% hemifusion inhibition ($^\#Fab_{1/2hemi}$) for H1N1 PR8 influenza virions.

the simulation, the observed yield may not be higher than simulated, and in general lower. In experiments at low Fab concentration, *Otterstrom et al. (2014)* reported as much as 55% fusion; with IgGs, up to 65% in individual measurements. Even without rescaling, both these values are higher than the simulated values of yield for $N_h$ = 3 or 4 at low Fab or IgG concentration. The more complete analysis in *Figure 8—figure supplement 2* rules out $N_h$ = 3 and disfavors $N_h$ = 4.

Simulation results for mean hemifusion delay, $N_{gamma}$ and $k_{gamma}$ remained relatively constant as a function of bound Fab for all $N_h$ (*Figure 8C–E*) because the corresponding $f_{un}$ was such that the starting point (no bound Fab) landed in the corresponding 'plateau' regions for these values (see *Figure 3*). Results for mean hemifusion delay times were indistinguishable for different $N_h$ and thus could not help discriminate among these various possibilities. Furthermore, the published data in *Otterstrom et al. (2014)* show relatively small (and hence noisy) samples for their H1N1 experiments (their Figure S8 and re-plotted here in *Figure 8*). As we show in *Figure 6—figure supplement 2*, estimates of $N_{gamma}$ from runs with only 100 particles scatter quite widely around the value used in the simulation, and the observed $N_{gamma}$ is thus not a good discriminator for deciding among $N_h$ values between 4 and 6. We conclude that for H1N1 PR8 viruses, $N_h$ is greater than 3 and might be higher than 4. A more precise estimate will require larger data sets. A consequence of the somewhat larger $N_h$ is that for H1N1 PR8 virions under the experimental conditions of *Otterstrom et al. (2014)*, the rate constant ($k_e$) for productive extension by individual HAs is ~0.034–0.035 sec$^{-1}$, nearly twice the rate of the corresponding step for H3 X31 influenza HA (see above).

## Discussion

The outcomes of simulations we report here and their application to analysis of newly published data on inhibition of fusion by stem-directed Fabs (*Otterstrom et al., 2014*) are fully consistent with the model developed in our previous papers (*Floyd et al., 2008*, *Ivanovic et al., 2013*). In that model, the number of HAs needed to generate a fusion event is not fixed by the organization of some intermediate state (e.g., by lateral interactions within a ring of HAs), but rather by the relationship between the free energy needed to overcome the kinetic barrier to hemifusion and the free energy gained in the HA$_2$ conformational transition. Variation in $N_h$ between influenza strains supports this mechanism. The new simulations extend the earlier model by including inactive (or inactivated) HAs and by showing that data on Fab inhibition can help restrict the estimates for the number of HAs required to generate hemifusion and the fraction of participating HAs.

Our new simulation results further expose limitations of the original analytical model that we and others used to interpret single-virion fusion kinetic data (*Floyd et al., 2008*, *Ivanovic et al., 2013*, *Otterstrom et al., 2014*). The standard analytical treatment of sequential kinetics (the gamma distribution) falls short, because the fusion mechanism involves stochastic events across a large enough interface that one of several potential initiating events will go on to completion. Even in the context of targeted HA inhibition analyzed here, and in a particular instance when most of the virions that are fusion competent have only a single potential region with $N_h$ active HA neighbors, the gamma distribution parameters, $N$ and $k$ do not reflect the underlying number of HA participants or the rate of their extension (*Figure 3*), because the $N_h$ HAs can extend in any order and there are more ways for the initial event to occur than for the next. Although the gamma distribution continues to be a useful tool to correlate experiment and simulation results, an updated analytical model would be needed to capture the fusogenic molecular events at the virus target-membrane interface, as we now understand them. Moreover, while our current simulation model does well in the context of accumulating single-virion membrane fusion data, it is likely that this model also will evolve as we gain new experimental insight. The experiment, computer simulation, and mathematical modeling will continue to evolve together, because they serve as independent tests for mutual validity and

reliability and because each can lead to predictions that can be tested by one of the other, complementary approaches.

We showed in our previous paper that the rate of fusion-peptide release from the pocket near the three fold axis sets the rate constant for target-membrane engagement (*Ivanovic et al., 2013*). This rate in turn depends on the overall stability of the pre-fusion conformation and (at a given pH) on the overall pK of the particular HA species. Simulations described here and comparisons with data from *Otterstrom et al. (2014)* identify two additional parameters that determine the overall rate of HA-mediated fusion — the number ($N_h$) of participating HA trimers required to distort the apposed membranes into a hemifusion stalk and the fraction within the contact zone of participating (active and productively refolded) HAs (*Figure 9*). We show by comparing data from an H3N2 strain and an H1N1 strain that $N_h$ can vary from one strain to another even under the same experimental conditions. (These differences may or may not represent subtype specific differences.) $N_h$ times the free energy recovered in the fold-back step from an extended intermediate to the postfusion 'trimer of hairpins' must exceed the kinetic barrier to hemifusion, estimated to be at least 50 kcal/mol (*Harrison, 2015*). It is reasonable to expect that the free-energy recovery, and hence the required $N_h$, will depend on the particular HA in question.

The fraction of participating HAs, which determines the probability that $N_h$ neighboring HAs will all be active, will depend on the percent of uncleaved $HA_0$, the percent of inhibitor-bound (e.g., Fab-bound) HA, and the probability that any particular HA will fail to engage the target membrane and instead fold back and insert its fusion peptides into the viral membrane. In addition to governing the rate of release, the fusion-peptide amino-acid sequence, which is very highly conserved (*Nobusawa et al., 1991*, *Cross et al., 2009*), may influence the efficiency of target-membrane insertion. It is also plausible that continued receptor engagement by $HA_1$ might contribute to the probability of target-membrane engagement (*Figure 1*). Ordering of HA conformational transitions in the context of membrane fusion may vary among strains, but some features are suggested by studies of soluble HA ectodomain (*Godley et al., 1992*, *Garcia et al., 2015*). If the C-terminus of $HA_2$ becomes disordered before the rest of the conformational changes that allow HA extension, then $HA_1$-receptor engagement will increase the probability that fusion peptide sequences project toward the target membrane instead of inserting back into the viral membrane (*Figure 1*).

The general approach developed in our previous papers (*Floyd et al., 2008*; *Ivanovic et al., 2013*) has also been used to study the flavivirus fusion mechanism (*Chao et al, 2014*). Flaviviruses have about 25% the surface area of even the smallest influenza virions and can display at most 60 trimers (about 15% of the number on a typical small influenza virus particle). A transition from dimer-clustered E-protein subunits to fusogenic trimers is a component of the mechanism not required when the fusogen is already trimeric like influenza virus HA. Nonetheless, the fusion mechanisms for the two groups of viruses are relatively similar. Trimerization of the flavivirus E-protein subunits and target-membrane engagement of their fusion loops are rate-limiting; hemifusion requires at least two adjacent trimers. Simulations show that trimerization is a bottleneck because of limited availability of competent monomers within the contact zone between virus and target membrane, so that trimer formation must await monomer activation (e.g., dimer dissociation). The basic concepts revealed by our current analyses might thus be generalizable to other viral membrane fusion systems.

The constraints imposed by fitting the hemifusion yield and the hemifusion delay time as functions of the number of Fab-inactivated HAs have allowed us to determine the fraction of unproductive HAs. This determination has in turn allowed us to associate the $k_{sim}$ value with the rate constant ($k_e$) for the limiting step during membrane engagement. Although the specific value for $f_{un}$ depended on patch size, the underlying rate constant did not (compare *Figure 9* and *Figure 9—figure supplement 1*). We have previously concluded from the fusion kinetics of HA mutants that the rate-limiting step of membrane engagement is the release of the fusion peptide from its 'prefusion' pocket near the threefold axis of the trimeric HA (*Ivanovic et al., 2013*). Reversible fluctuations at $HA_1$:$HA_1$, $HA_1$:$HA_2$ and $HA_2$:$HA_2$ interfaces (*Figure 1*, 'open state') determine a 'window of opportunity' for fusion-peptide release and for their irreversible projection beyond the outer margins of adjacent $HA_1$ heads. The link between $k_e$ and the rate constant for a specific molecular rearrangement should in the longer term allow us to derive direct information for individual fusion catalysts in a functionally relevant context.

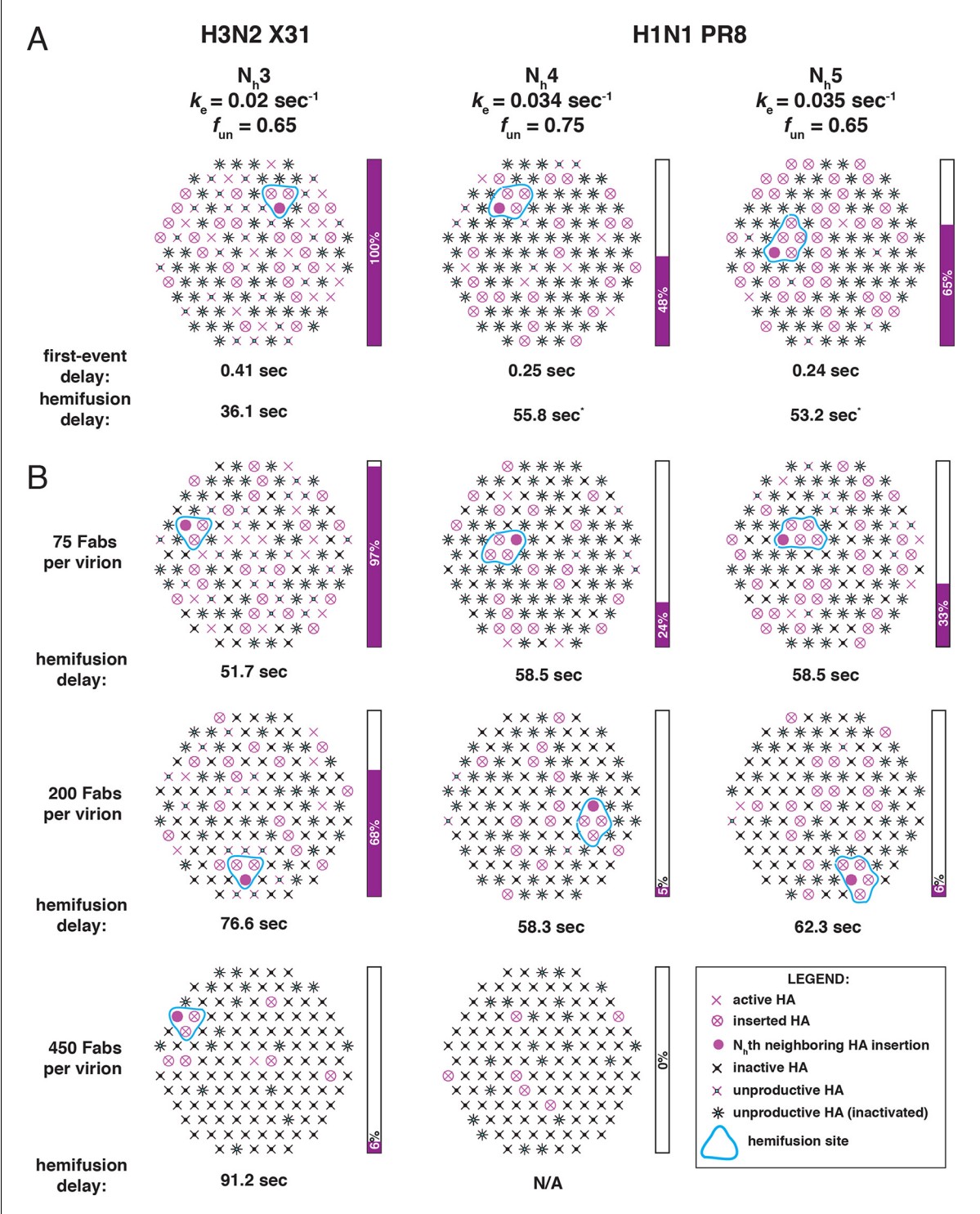

**Figure 9.** Independent functional determinants of HA-mediated membrane fusion and their effects on the influenza virus susceptibility to neutralization. Conclusions are presented in the context of the PS = 121 contact patch. (**A**) The rate of irreversible HA extension ($k_e$) and the frequency of unproductive or inactive HAs determine the rate of target membrane engagement by individual HAs. First-event delay – the average time to the first HA conversion, either productive or non-productive – is determined solely by the $k_e$ and the patch size. (See *Figure 9—figure supplement 1* for the corresponding model that uses PS = 55). Stochastic HA triggering dictates that small changes in the number ($N_h$) of HAs required for fold-back have significant effects on the kinetics of fusion. Small increases in $N_h$ significantly reduce the extent of fusion (*purple bars*) in the context of the large $f_{un}$ values. Compensatory differences in $k_e$, $f_{un}$ and $N_h$ between X31 H3N2 and PR8 H1N1 influenza result in similar overall rates of hemifusion (delay of about 36 and 58 sec, respectively). [*]Note that by exchanging the $k_e$ values between the H3 and H1 functional variables (i.e. compare results for $k_e = 0.034$ sec[-1], $N_h = 3$,
*Figure 9 continued on next page*

*Figure 9 continued*

$f_{un}$ = 0.65 and $k_e$ = 0.02 sec$_{-1}$, $N_h$ = 4 or 5, $f_{un}$ = 0.75 or 0.65), we obtain 'extreme' values for hemifusion delay or ~20 and ~100 sec, respectively. (**B**) Illustration of the effects of Fab binding on fusion kinetics (mean hemifusion delay) and the theoretical hemifusion yield (*purple bars*) in the context of functional variables revealed for H3N2 X31 and H1N1 PR8 influenza virions. Our conclusions reveal an intricate link between the molecular features of the evolved fusion mechanism and its susceptibility to neutralization.

The following figure supplement is available for figure 9:

**Figure supplement 1.** Independent functional determinants of HA-mediated membrane fusion and their effects on the influenza virus susceptibility to neutralization.

In our analysis of conformational changes for virion-associated HA in the absence of target membranes, we have made the unexpected observation that the rate of irreversible inactivation for X31 HA is accelerated at the target membrane interface. It took 10 min at pH5.2 and 37C for about half of the HAs on a virion surface to inactivate irreversibly (*Figure 4* and *Figure 4—figure supplement 1*). The same virions hemifuse with a mean delay of ~1 min at the same pH and at a much lower temperature (23°C) (*Ivanovic et al., 2013*). According to our current simulation model, for $f_{np}$ = 0.5 and $N_h$ = 3, at the time of hemifusion, an average of 34% of HAs at the target-membrane interface are no longer in the pre-fusion conformation (i.e. have inserted in the target membrane or become inactivated). This is at least an order of magnitude greater than their rate of inactivation on free virions. Because the frequency of non-productive HA refolding is high (at least ~50%), the presence of a target membrane appears to accelerate both productive and non-productive refolding. We illustrate in *Figure 1* a model that could explain these observations. Receptor engagement might retain HA$_1$ in a configuration separated from HA$_2$ (an 'open-HA" conformation) and thereby increase the time interval for fusion peptide release and irreversible HA extension. Receptor engagement might also influence the ratio of membrane insertion to HA inactivation (see our earlier comment), but an overall increase in the rate of committed HA extension would in any case increase the rate at which HAs reach one or the other of those endpoints. The degree of rate increase (with respect to inactivation of HAs on free virions) will depend on the relationship between the lifetime of the open state and the probability of fusion-peptide release during the interval when HA$_1$ is not in the way. Udorn HA does not exhibit the same relative increase in the rate of refolding (*Figure 4* and *Figure 4—figure supplement 1*). After 1 minute of incubation at low pH, most of its virion-associated HAs have assumed the low-pH conformation, but the rate of Udorn hemifusion at pH 5.2 is only ~twofold higher than that of X-31 (*Ivanovic et al, 2013*). Udorn HA, with a destabilized docking of the fusion peptide, appears to have a much greater probability of fusion-peptide release during its unconstrained (i.e. on free virions) open-state lifetime than does X-31 HA, which requires, for comparably rapid extension, the increased open-state lifetime afforded by receptor interactions with HA$_1$. The Udorn fusion peptide might, however, be less efficient at inserting into the target membrane, because of the mutation of Gly to Ser at its fourth position. If so, the ratio of non-productive to productive HA transitions might be higher for Udorn than for X-31. The proposed role for HA-receptor contacts in catalysis of membrane fusion, not just in cell attachment, should be directly testable by future single-virion membrane fusion experiments. An important consequence of this possibility is that adjustments in receptor affinity would effectively modulate not only the yield and kinetics of fusion, but also the susceptibility of the virus to neutralization (*Figure 9B*).

The rate of fusion-peptide exposure is higher for HA from PR8 H1N1 virus than for HA from X-31 H3N2, but a greater $N_h$ and potentially also a decreased productivity of refolding for the former strain leads to a somewhat lower overall rate of fusion (panel A in *Figure 9* and *Figure 9—figure supplement 1*). Thus, compensatory changes appear to maintain the overall rate within an acceptable range and imply some degree of independence of the molecular mechanisms that modulate the three fusion-rate determinants. Influenza virus penetrates from low-pH endosomes, and the rate of fusion may have an optimum determined by a balance between the rate of acidification of the virion interior (required to release viral RNPs from the matrix protein [*Martin and Helenius, 1991*]) and the efficiency of penetration before the virus particle undergoes lysosomal degradation (*Ivanovic et al., 2012*). Replication of influenza virus in birds, humans and pigs is constrained by different kinds of pressures on its cell-entry machinery (stability of HA in the extracellular environment

and its roles in receptor binding and membrane fusion) (*Schrauwen and Fouchier, 2014*). Distinct mechanisms that independently modulate the properties of this molecular machinery might determine the potential of a given strain to adapt to replication in a new host. Similar considerations will determine the potential of HA to evolve resistance to inhibitors that target it.

Higher $N_h$ (combined with relatively low productivity of HA refolding) reduces the baseline yield of fusion and increases the susceptibility of the H1N1 strain used by *Otterstrom et al. (2014)* to a fusion inhibitor (antibody) (*Figure 9B*). A recent study of HIV-1 cell entry combined experiment and simulation to show infectivity differences among HIV-1 strains that differ in the number of participating fusion proteins required for entry (*Brandenberg et al., 2015*). Further studies of the range over which $N_h$ can vary among influenza strains, even within subtypes, and molecular determinants of $N_h$, will be valuable for assessing levels of antibodies (or other entry inhibitors) required for protection.

The high percentage of unproductive HAs is probably the most unexpected result of our analysis. In our own experiments, cleavage was complete, so remaining $HA_0$ is not the reason for this observation. After release of the fusion peptide and formation of an extended intermediate (driven, presumably, by the strong $\alpha$-helical propensity of the segment between the $\alpha$1 and $\alpha$2 helices in $HA_2$: *Carr and Kim, 1993*), the relative efficiency of membrane engagement, which traps the extended intermediate, and $HA_2$ fold-back will determine whether the HA is productive or not. Under the conditions of our experiments (*Floyd et al., 2008*, *Ivanovic et al., 2013*) and those of *Otterstrom et al. (2014)*, the two efficiencies appear to be comparable, and fusion occurs even with more than half of the HAs inactive. The relatively large proportion of non-productive conformational transitions ($f_{un}$ ~0.65-0.75) (*Figure 9*) lies within the region of the fusion inhibition curve in which small changes in $f_{un}$ will influence both yield and rate (see *Figure 3*). The large effect on fusion of a small number of bound antibodies (*Otterstrom et al., 2014*) is consistent with this prediction. A potential evolvability benefit for the virus is that a small decrease in $f_{un}$ will have a comparably strong effect, directly offsetting the effects of antibodies or potential fusion inhibitors. The relative insensitivity of the fusion mechanism to a high ratio of unproductive to productive HAs, and the potential for a direct contribution to the efficiency of fusion from adjustments in the fraction of non-productive events, combine to produce an extremely robust general mechanism.

## Materials and methods

### Virions
Strains
Virions used by *Floyd et al. (2008)* were A/Aichi,X31/2/68(H3N2). The HA open reading frame from that virus stock was reverse transcribed and used to generate X31HA-Udorn virions by replacing Udorn-HA open reading frame in reverse genetics constructs for A/Udorn/72 (H3N2) (*Ivanovic et al., 2013*). That study also used WT A/Udorn/72 (UdornHA-Udorn) virions and a number of HA mutants in either background. Virions used by *Otterstrom et al. (2014)* were A/Aichi,X31/2/68(H3N2) and A/PR/8/34 (H1N1), designated as X31 and PR8, respectively.

Patch size
We previously estimated that a spherical influenza virion with a 55 nm membrane-to-membrane diameter incorporates about 50 HAs in its target-membrane contact patch (*Ivanovic et al., 2013*) (*Figure 1B*). Egg-derived X31 virions used by *Floyd et al. (2008)* were mostly spherical particles of this size. X31HA-Udorn virions and UdornHA-Udorn preparations used by *Ivanovic et al. (2013)* were enriched in slightly elongated particles with membrane-to-membrane distances of about 130 × 55 nm, and their contact patch was estimated to include about 120 HAs. X31 and PR8 virions used by *Otterstrom et al. (2014)* appeared as larger spheres in electron micrographs, with average diameters of about 125 nm, probably because of rounding and flattening in the uranyl acetate stain. Influenza virions lose their filamentous morphology at low pH (*Calder et al., 2010*), and we found similar effects when using uranyl acetate. Because of this ambiguity, we included two patch sizes (PS), 121 and 55, in all simulations and comparisons with data in *Otterstrom et al. (2014)*, but we found that the fundamental conclusions derived from the current analysis are independent of the patch size. We show simulation results for PS = 121 as main figures and those for PS = 55 as figure supplements.

## Computer simulation

We used the computer simulation algorithm we described previously (*Ivanovic et al., 2013*) with several modifications indicated below and annotated in the accompanying Source code (the script – s_arrest_hemifusion_simulation_eLife2015resubmission.m, and the functions used by the script – generate_patch.m, s_randomdist.m, isaN2tuplet6AllGeos.m, and findFlippedNeighbors.m). In brief, we defined a circular contact patch incorporating either 121 or 55 HAs arranged in a hexagonal lattice, where each internal HA has exactly 6 HA neighbors (*Figure 2A*). For simulations involving virions with a reduced fraction of active HAs, a defined fraction of HAs in random positions within the contact patch were flagged as inactive or unproductive (different random positions for each analyzed virion) (*Figure 2D*). We assumed a single-step process for the irreversible extension of individual HAs leading either to membrane insertion (active HA; productive path, *Figure 1A*) or inactivation (unproductive HA; non-productive path, *Figure 1A*). We first derived lag times for each HA (both active and inactive/unproductive) in the contact patch by random drawing from an exponentially decaying function with rate constant, $k_{sim}$ (see below). We then sorted these times in ascending order and defined hemifusion time as the lag-time for the *active* HA that contributed the final, $N_h$th member to the previously inserted group of ($N_h$-1) *active* HA neighbors. Inactive or unproductive HAs simply could not contribute to the inserted HA neighborhood. If the hemifusion event was not detected after the HA with the longest lag time was analyzed, the given 'virion' was flagged as 'dead'. The simulation process was repeated as many times ($n_{total}$) as needed to yield ~1000–3000 hemifusion events ($n_{hemi}$) for all results shown. We defined hemifusion yield as $100(n_{hemi}/n_{total})$.

## HA neighborhoods

We previously defined $N_h$ = 3–5 neighborhoods (*Ivanovic et al., 2013*) and illustrated them again here (*Figure 2*). We have now extended the code to include a possibility of fusion-inducing HA sixmers, a group of 6 HA neighbors that might cooperate during fold-back (*Figure 2C* and *Figure 2—figure supplement 1*; modified function, 'isaN2tuplet6AllGeos.m', is submitted as Source code).

## Fab inhibition and unproductive HAs

We assumed that Fabs bind randomly to HA monomers. The fractional monomer occupancy by a given number of Fabs (#Fab) was therefore #Fab/3*#HA. We used 375 as the number of HAs per virion (#HA) (estimate based on cryoEM data of spherical influenza virions: *Harris et al., 2006*, *Calder et al., 2010*, *Wasilewski et al., 2012*). The frequency of HAs with no bound Fab was (1-monomer occupancy)[3]. This value represents the participating HA fraction for 100% productivity of HA refolding. In considering two different patch sizes, the same total number of HAs was used for estimates of frequency of unbound sites.

For simulations in which we considered reduced productivity, we included as inactive an additional fraction of the Fab-free sites. We combined the frequency of Fab-bound ($f_{Fab}$) and unproductive ($f_{un}$) HAs in the common factor, frequency of non-participating HAs ($f_{np}$) as follows: $f_{np}$ = $f_{Fab}$+ (1- $f_{Fab}$) * $f_{un}$. We calculated this $f_{np}$ value and entered it into the original code (*Ivanovic et al., 2013*) as the fraction inactive HAs. We subsequently updated the code (see Source code) to allow entry of separate values for $f_{Fab}$ and $f_{un}$, treating them in a manner analogous to what we did manually to derive illustrations shown in *Figures 2* and *5–9*. In *Figures 2*, *6*, *8* and *9*, unproductive HAs are shown as inactivated if their times of inactivation preceded the time of hemifusion.

## $k_{sim}$

We have adjusted the values for $k_{sim}$ from those in our original study (*Ivanovic et al., 2013*) to match the hemifusion lag times measured by *Otterstrom et al. (2014)* at pH 5, while taking into account the new interpretation that a large portion of the HA molecules in contact with the target inactivate irreversibly. The value for $k_{sim}$ we used originally, 0.0025 sec[-1], yields a mean hemifusion delay of ~75 sec (pH-drop to hemifusion, see next paragraph), closely matching those previous experiments (at ~pH5.2-5.5) if we assume that all HAs in the contact patch can contribute to fusion (*Ivanovic et al., 2013*). To yield a mean hemifusion delay of ~75 sec when 65% of HA molecules in the contact patch are unproductive (*Figure 9*, X31 H3 panel), a higher value for $k_{sim}$ is required (0.0095 sec[-1]). So, in the final model (*Figure 9* and *Figure 9—figure supplement 1*) for H3 simulations we used $k_{sim}$ = 0.02 sec[-1] (yielding a mean of ~36 sec or a geometric mean of ~30 sec), and for

H1 simulations we used $k_{sim}$ = 0.034 or 0.035 sec$^{-1}$ (yielding the mean of ~56 sec or the geometric mean of ~47 sec) (*Figure 6*, *8* and *9*). (Compare also the simulation-derived mean hemifusion delay for the H3 strain (*Figure 9*) to that shown in *Figure 2B*, which uses the same $k_{sim}$ value but $f_{np}$ = 0). Increasing the value for $k_{sim}$ decreases the mean lag time to hemifusion and $k_{gamma}$ without affecting any of the parameters derived and plotted in *Figure 3*: hemifusion yield, mean hemifusion delay normalized to $f_{np}$ = 0, $N_{gamma}$, or the $k_{gamma}/k_{sim}$ ratio.

### $N_{gamma}$ and the arrest intermediate

All current simulations-derived delay times reported the time from pH drop to hemifusion, to facilitate comparison with previous experiments (*Floyd et al., 2008*, *Otterstrom et al., 2014*). The only previous exceptions were our experiments that used X31HA-Udorn virions and related UdornHA-Udorn mutants (*Ivanovic et al., 2013*), which were mobile at pH drop and for which a separate, arrest intermediate was considered (times when virions stopped moving). In those cases, published delays reflected separately times from pH drop to virion arrest and times from virion arrest to hemifusion. To compare current simulation results with the previous experimental data, we determined $N_{gamma}^{(pH\ drop\ to\ hemifusion)}$ ($N$ value derived from fitting pH drop to hemifusion lag-time frequency distributions with the gamma probability density), for those published datasets (*Figure 3—figure supplement 3*).

Simulation results for $N_{gamma}$ show significant scatter for smaller sample sizes ($\leq$100 events) (*Figure 6—figure supplement 2*). As a result, we rely more on previous measurements of $N_{gamma}$ from larger sample sizes (at least 400 virions) in our various analyses. *Floyd et al. (2008)* reported $N_{gamma}$ values between 2.7 and 3.4 for spherical (PS = 55) H3 X31 virions (n = 450–1080). *Figure 3—figure supplement 3* shows these values for slightly elongated (PS = 121) X31HA-Udorn, UdornHA-Udorn and their point mutants, X31HA$^{G4S}$-Udorn and UdornHA$^{S4G}$-Udorn virions (n = 409–970).

## Virion-HA processing and low-pH conversion experiments

A2 antibody hybridomas were a generous gift from Judith White, University of Virginia. LC89 antibody was a generous gift from Stephen Wharton, MRC National Institute for Research, London, UK.

We previously verified that HA was completely processed to HA$_1$:HA$_2$ on all virions that were used in *Ivanovic et al. (2013)* study. We show this result here for WT virions of two different X31HA-Udorn and UdornHA-Udorn virus preparations used in that study (each was derived from a separate plaque during initial purification). We further demonstrate the ability of these virion-associated HAs to convert to their low-pH form (*Figure 4* and *Figure 4—figure supplement 1*).

### Western blots

All samples were separated on 8% SDS-polyacrylamide gels and transferred onto a 0.45-μm PVDF membrane and probed with A2 antibody specific for HA$_1$ (*Copeland et al., 1986*).

### HA processing

Purified virions were stored in virion buffer (20 mM Hepes-NaOH pH7.4, 150 mM NaCl, 1 mM EDTA). Stock concentrations were normalized based on absorbance at 280 nm (A$_{280}$ ~4) and an equivalent of 0.4 μl per sample of an appropriate virus dilution was loaded per each virion lane. About 100 ng of purified recombinant X31 HA$_0$ or HA$_1$:HA$_2$ was loaded as a reference.

### Low-pH conversion

2.5 μl of normalized virus stocks were diluted with 47.5 μl of low-pH buffer (10 mM citrate pH5.2, 140 mM NaCl, 0.2 mM EDTA) and incubated in a 37C water bath for indicated times before neutralization with 5 μl neutralization buffer (750 mM Tris-HCl pH7.5). Neutral-pH samples were directly mixed with 52.5 μl reneutralized buffer (47.5 μl pH 5.2 buffer pre-mixed with 5 μl neutralization buffer). At indicated time points reneutralized and neutral samples were split into equal aliquots (0.4 μl original virus stock equivalents). One aliquot was loaded directly onto the gel (no trypsin samples in *Figure 3B* or input samples in *Figure 3C*). A second aliquot was treated with trypsin (40 μg/ml final trypsin concentration) for 30 min on ice (trypsin was inactivated using 0.5-1mM PMSF). The remaining aliquot was subjected to immunoprecipitation (IP) with conformation-specific anti-HA antibody, LC89 (specific for the low-pH form of HA, HA$_2$ epitope (*Wharton et al., 1995*)), as follows.

Samples were mixed with ~1.4 μg (LC89) antibody in a 12 μl reaction additionally consisting of 1% NP40 and incubated overnight at 4C. Protein G beads (Dynabeads, Life Technologies) were washed twice with IP buffer (10 mM Tris-HCl pH 7.4, 150 mM NaCl, 1% NP40) and then resuspended in this buffer. An equivalent of 10 μl beads was added to each IP sample in 8 μl total buffer volume. Bead-containing reactions were incubated for 2 hours with gentle rotation at 4C. Total IP supernatant was mixed with reducing gel sample buffer and boiled. IP pellet (beads) were washed three times with IP buffer, then resuspended in reducing gel sample buffer and boiled. Total IP pellet and supernatant were loaded onto gels.

## Acknowledgements

We thank Antoine van Oijen for comments on the manuscript. We thank Stephen Wharton and Judith White for reagents. SCH is an investigator of the Howard Hughes Medical Institute.

## Additional information

### Competing interests

SCH: Reviewing editor, *eLife.* The other author declares that no competing interests exist.

### Funding

| Funder | Grant reference number | Author |
| --- | --- | --- |
| National Institutes of Health | U54AI057159 | Tijana Ivanovic<br>Stephen C Harrison |
| Howard Hughes Medical Institute | | Stephen C Harrison |

The funders had no role in study design, data collection and interpretation, or the decision to submit the work for publication.

### Author contributions

TI, Conception and design, Acquisition of data, Analysis and interpretation of data, Drafting or revising the article; SCH, Analysis and interpretation of data, Drafting or revising the article

## Additional files

### Supplementary files

• Source code 1. The simulation model of fusogenic molecular events at the virus target-membrane interface. The code consists of the main code text and four functions written in MATLAB (version R2015a). The main code script is titled s_arrest_hemifusion_simulation_eLife2015resubmission.m, and the functions are generate_patch.m, s_randomdist.m, isaN2tuplet6AllGeos.m, and findFlippedNeighbors.m. The simulation process is outlined within the main code text and in the Computer Simulation subsection of the Materials and methods. The code was adapted from Ivanovic et al (2013) to include a possibility of $N_h=6$ and the unproductive HA population, and to measure hemifusion delay from the start of the simulation rather than from the arrest intermediate.

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
