## [Decision Letter]

Thank you for submitting your work entitled "Distinct kinetic determinants of influenza hemagglutinin-mediated membrane fusion" for peer review at *eLife*. Your submission has been generally favorably evaluated by Michael Marletta (Senior editor), a Reviewing editor (Axel Brunger), and three reviewers, but they identified several areas that require some revision.

The reviewers have discussed the reviews with one another and the Reviewing editor has drafted this decision to help you prepare a revised submission.

Summary:

This manuscript presents a quantitative analysis of influenza single particle membrane fusion data that represents an important contribution to understanding this process at the molecular level. The authors have previously developed a simulation model for these single virion fusion studies and in this manuscript substantially extend this analysis to gain insight into the dependency of experimental observables (e.g. fusion delay times and fusion yield) as a function of non-active HA trimers, the number of HA trimers required for fusion and the number of antibody Fabs that inhibit fusion. The parameters that govern the overall process of fusion and its susceptibility to inhibition by Fabs suggest why some influenza strains may be more easily neutralized by antibody and how physical attributes of the individual HA trimers can influence both fusion and neutralization. The simulation model also makes specific predictions about individual HA trimer activation rates and contributions to the free energy requirements to drive membrane fusion that are likely to be testable in future experiments.

Essential revisions:

1) The current manuscript is quite dense and takes significant effort to work through many details of both the model and the simulation data presented. The authors are encouraged to simplify some of the figures and to divide some of these further (e.g. Figure 1) so that the main points can be more easily followed by a reader who is unfamiliar with the approach. Specific suggestions by one of the reviewers are provided below. Some of the detailed graphs could be provided as supplementary data figures.

2) The predicted N from the gamma model is dependent on the fraction of non-productive HA trimers (quite striking in Figure 2). However, one would not expect the number of required active trimers to trigger fusion be dependent on the fraction of non-productive or antibody-inactivated HA trimers. Thus, the gamma distribution model is indeed questionable if there is a significant fraction of non-productive "molecules". Perhaps this limitation of the gamma function treatment should be made more explicit since it has rather general implications (as hinted in the manuscript).

3) Although the simulation model seems physically more reasonable than the gamma distribution model, the derived parameters (*k, N*, etc.) are not entirely unique as discussed in this work. Moreover, the simulation model itself may not be unique. Future experimental data may require further revision of the model or even perhaps a different model. Some discussion of this point might be appropriate.

4) Cooperativity of the action of fusion proteins in conjunction with the membrane has been inferred from the simulation model analysis of kinetic traces. However, is there direct evidence for the arrest to be in an extended conformation prior to HA fold-back?

5) Paragraph two, subheading “Simulations of molecular events at the virus-target membrane interface”: how do these increases in time to hemifusion compare to those from the Otterstrom data?

6) Paragraph three, subheading “Simulations of molecular events at the virus-target membrane interface” regarding lag time plateaus: is there an increase in bound virus that does not fuse when the hemifusion lag plateau is reached?

7) The fusion kinetics for X31 and Udorn are faster than the 10-minute incubation times shown in the immunoblots. It would be helpful to see time courses with time points better corresponding to fusion timescales.

8) The simulation model uses the parameter *k*, the rate of activation of independent HA. In the present manuscript, a somewhat different rate is used compared to the 2013 *eLife* work where *k* = 0.0025 /s for all simulations of X31 (H3N2) influenza. Here *k*_sim_ was changed so that it matched experiments in Otterstrom 2014 (paragraph two, subheading “Fab inhibition of H3 HA” and paragraph two, subheading “Fab inhibition of H1 HA”). Question: is the resulting fit rather insensitive to changes of 5-10 in this rate? This point may need some discussion.

9) Abstract: comparing only 2 strains and one mutant might be too few to make a general statement about HA evolvability. Perhaps this statement should be softened.

Specific suggestions on the figures:

Figure 1 should be revised to represent each of the parameters of the modeling more clearly. A schematic of the model that explicitly shows how each of the main parameters (N_h_, *k*_sim_, *f*_np_, antibody Fab, and patch size) plays a role in the process of fusion is important. Figure 1 shows prefusion productive vs. non-productive HA refolding, but a more comprehensive schematic overview of all of the key steps that are being modeled would be useful in order to quickly grasp what is being simulated.

Leave out the HA structural details in an overall schematic of the modeling, although it is still useful for showing productive vs. non-productive HA.

Use a simple representation for active HA trimers, inactive HA trimers, or Fab-inhibited trimers.

Make the schematic multiple panels, displaying the HA molecules (active and inactive) at the interface of the viral and cellular membranes.

Within each panel one can highlight/define two model parameters – the fraction of non-participating trimers (*f*_np_) and the patch size (PS).

Use multiple panels to show individual HA trimers being activated (to highlight/represent *k*_sim_) and also use the number of panels to indicate the number of activated HA trimers required for driving membrane fusion (e.g three activation panels for N_h_ of 3).

There could be an indicator for the overall fusion delay time above the panels that helps tie this observable to the discrete steps affected by *k*_sim_, *f*_np_ and N_h_

Split Figure 1 into two figures. The first could show the hemifusion and hemifusion delay predictions and the second could show the *N*_gamma_ and *k*_gamma_. There may be other ways to divide these up (e.g. separated by patch size).

In Figure 5, consider putting the PS55 panels into supplementary information. The PS121 are really the most interesting with comparisons to the experimental data.

Figure 4 and Figure 6: Although the figure has boxes around the parameter sets of N_h_ and *f*_np_ that agree best with the Fab inhibition data, the main point is still mostly buried. The figures could be simplified somehow, by de-emphasizing the majority of the bars that are not relevant to the main point being made (e.g. using different colors or some other approach).

Figure 7: Same as Figure 5. Consider moving the left panels to supplementary figures and just show the predictions with the experimental data. The authors should discuss why they think the experimentally observed hemifusion delay could increase as a function of the bound Fabs in contrast to the model.

Figure 8: panel A could be removed if a comprehensive schematic is shown in Figure 1. In this current panel 8A the model parameters are insufficiently represented. *f*_np_ is shown as an arrow, similar to k_fp_ (or *k*_sim_). The participating HA trimers go immediately to two postfusion trimers, indicating N_h_. However, this is too similar to a standard representation of fusion that loses the key point that there are multiple independent HA activation events contributing to the accumulation of N_h_ trimers, which can then drive fusion. Panel 8B also could use some revision to make the H3N2 and H1N1 comparison more clearly. Question: what do the filled diamonds represent?

---

## [Author Response]

Essential revisions:1) The current manuscript is quite dense and takes significant effort to work through many details of both the model and the simulation data presented. The authors are encouraged to simplify some of the figures and to divide some of these further (e.g. Figure 1) so that the main points can be more easily followed by a reader who is unfamiliar with the approach. Specific suggestions by one of the reviewers are provided below. Some of the detailed graphs could be provided as supplementary data figures.

We have complied with all the suggestions for modifying the figures as outlined here and in our responses to the reviewer’s specific suggestions below. We have also made textual changes to simplify the description of the simulation analyses and to improve the manuscript flow. Key textual changes are highlighted in red in the tracked-changes version of the manuscript text (figure legends were substantially modified to reflect figure changes and reshuffling but were not highlighted unless in a reference to a specific question below).

Figure revisions:

1) All PS55 panels have now been moved to figure supplements. The description of the simulation results now primarily focuses on the larger patch size, PS121. PS55 panels are now referenced only in a few places in the main text without emphasizing detail, but rather focusing on general trends and conceptualizing results for the smaller patch size as a unique instance of a larger patch size with a smaller number of participating sites. See the Results section (subsections “Simulations of molecular events at the virus-target membrane interface”, “Fab inhibition of H3 HA”, and “Fab inhibition of H1 HA”) and the Materials and methods subsection, “Patch size”.

2) We have included additional detail in the diagram of HA conformational changes leading either to productive or non-productive HA refolding (Figure 1) with two main goals: 1) to better illustrate a model for the extended state of HA (your revision comment 4 below) and 2) to facilitate discussion of the proposed active role of HA-receptor contacts in membrane-fusion catalysis (paragraph seven, Discussion). This detailed image has also allowed us to remove a similar diagram from the model figure at the end (per your specific suggestion below).

3) We have added a 3D illustration of our fusion model (Figure 1, adapted from our 2013 *eLife* publication) to facilitate its description in the Introduction and to conceptualize diagrams of contact patches we went on to introduce in the new Figure 2.

4) We have added a new Figure 2, which illustrates all the simulated fusion parameters using diagrams of contact patches.

5) To all the simulation figures, Figure 3, Figure 5, Figure 6, Figure 7, Figure 8 and Figure 9, and their corresponding PS55 supplements, we have added illustrations of what is being simulated using patch-size diagrams introduced in the new Figure 2.

6) We have introduced a new parameter (*f*_un_ – fraction unproductive HAs – to replace *f*_np_ in several instances where its use might have been confusing previously). We introduce this parameter at the end of the Results section titled “Evidence for non-productive HA refolding”, and explain in the Methods under “Fab inhibition and unproductive HAs” how this fraction combines with the frequency of antibody-bound HAs to yield the value for *f*_np_ used in simulations that generated Figure 5-8.

7) We have generated a new model Figure 9. The main mechanistic conclusions are now summarized in panel A using patch-size diagrams to illustrate both N_h_ and *f*_un_. We use vertical bars to illustrate the theoretical fusion yield. We show mean first-event delays to illustrate the effect of *k*_e_, which we contrast with the overall hemifusion-delay values just below them. Panel B now illustrates consequences of the evolved fusion mechanistic features for the two analyzed strains on their susceptibilities to neutralization.

8) We have simplified both Figure 6—figure supplement 3 and Figure 8—figure supplement 2: our tests of sensitivity of the key conclusions to potential uncertainty in the measurements for the number of antibodies needed for hemifusion inhibition, and we have removed PS55 panels altogether and simplified their descriptions in the figure legends.

2) The predicted N from the gamma model is dependent on the fraction of non-productive HA trimers (quite striking in Figure 2). However, one would not expect the number of required active trimers to trigger fusion be dependent on the fraction of non-productive or antibody-inactivated HA trimers. Thus, the gamma distribution model is indeed questionable if there is a significant fraction of non-productive "molecules". Perhaps this limitation of the gamma function treatment should be made more explicit since it has rather general implications (as hinted in the manuscript).

Thank you for this suggestion. We have added a paragraph in the Discussion to make the statement of the limitations of the gamma function more explicit (paragraph two).

*3) Although the simulation model seems physically more reasonable than the gamma distribution model, the derived parameters (*k, N*, etc.) are not entirely unique as discussed in this work. Moreover, the simulation model itself may not be unique. Future experimental data may require further revision of the model or even perhaps a different model. Some discussion of this point might be appropriate.*

We agree with the general comment and have added a statement in the Discussion (paragraph two) suggesting that the simulation model is likely to evolve as new data are gathered. We have also added the description of a couple of new research directions this work will lead us into (see Discussion, paragraphs seven and ten). However, we feel that current data strongly favors values for certain fundamental parameters despite some remaining uncertainty. While it is correct that the exact value for the fraction of non-participating sites is patch-size dependent, the values for the number (N_h_) of participating HA neighbors and the derived rate (*k*_e_) of membrane engagement are patch-size independent, and, thus, unique for the current simulation model. We have revised the text to make that clear – see for example the Results (paragraphs two and three, subheading “Fab inhibition of H3 HA” and paragraph two, subheading “Fab inhibition of H1 HA”) and the Discussion (paragraph six). It is the differences between the two strains, rather than the specifics of the particular numbers, from which we draw the majority of our conclusions. For example, the different values for N_h_ for the H1 and H3 strains support the ‘tug-of-war’ model of membrane fusion (see Discussion, paragraph one). Further, differences in independent functional variables provide evidence for compensatory features in the evolved mechanism (see Discussion, paragraph eight) and establish a direct link between the fusion mechanism and viral neutralization susceptibility (see Figure 9 and the Discussion section).

4) Cooperativity of the action of fusion proteins in conjunction with the membrane has been inferred from the simulation model analysis of kinetic traces. However, is there direct evidence for the arrest to be in an extended conformation prior to HA fold-back?

This question relates back to our previous *eLife* 2013 publication, upon which this manuscript builds. In that prior paper we provided evidence that the arrest results from target membrane insertions of several HAs anywhere in the area of target-membrane contact. At that time, we also discussed that in order for fusion peptides to reach the target membrane and insert, they need to extend beyond the outer margins of adjacent HA1 heads (see also the Discussion section, paragraph six). While there is no direct information for the specific molecular structure of the ‘extended state’, in the revised Figure 1, we now illustrate two general ways this extra length could be gained. The extension of the N-terminal helical coiled-coil in HA2 that projects the fusion peptides toward the target membrane is essential, while the C-terminal portion of the molecule could either retain the pre-fusion conformation (the first extended-HA intermediate in Figure 1) or unfold to allow the onset of fold-back (the second and third extended-HA intermediates). *The key point is that rather than representing a unique conformation, the extended state is likely an ensemble of folded-back conformations*. We have added a sentence in the Introduction to reflect this point (paragraph three). We have also added a statement in the revised Figure 1 legend that the corresponding distance between the two membranes linked by inserted HAs might fluctuate around a different set of values depending on how many HAs are cooperating.

5) Paragraph two, subheading “Simulations of molecular events at the virus-target membrane interface”: how do these increases in time to hemifusion compare to those from the Otterstrom data?

Thank you for this question. This was a good opportunity to compare our initial simulation results (current Figure 3) to experiment data reported by Otterstrom et al. (2014) and draw some preliminary conclusions. In the Results, paragraph two, subsection “Simulations of molecular events at the virus-target membrane interface”, we have now commented on their observations of the effects of targeted HA inhibition on both the fusion yield and increases in time to hemifusion. The difference between the theoretical predictions and their experiment data suggested to us that in addition to HAs inactivated by antibodies, there are HAs that abort anyway (are unproductive). This initial comparison might also improve the flow of our analyses in reference to your revision-point 1 above by setting the stage for subsequent detailed comparisons with their data in Figure 5, Figure 6, Figure 7, Figure 8 and Figure 9.

6) Paragraph three, subheading “Simulations of molecular events at the virus-target membrane interface” regarding lag time plateaus: is there an increase in bound virus that does not fuse when the hemifusion lag plateau is reached?

Yes, for antibody/Fab concentrations in the plateau region for hemifusion delay (≥100 to 1000 nM), Otterstrom et al. found a continued decrease in hemifusion yield as antibody/Fab concentrations increased. We have added this observation in the Results (please see “Simulations of molecular events at the virus-target membrane interface”).

7) The fusion kinetics for X31 and Udorn are faster than the 10-minute incubation times shown in the immunoblots. It would be helpful to see time courses with time points better corresponding to fusion timescales.

The purpose of this figure was to show that all HAs on virions we had used in fusion experiments (*eLife*, 2013) have the potential to convert to the low pH state, which they do (see Figure 4 and Figure 4–figure supplement). However, the kinetics of inactivation of HAs on free virions (at least for X31) is significantly slower than in the context of membrane fusion, and this is the reason for including longer time courses for the virion-HA inactivation experiments (shorter time courses for X31 virions would not be informative). We did, however, in response to your suggestion, include an earlier time point for Udorn, which exposes an interesting difference between the two HAs and suggests a model that explains both the slow inactivation of X31 HA on free virions and the differential acceleration of X31 vs. Udorn HAs in association with target membranes. The proposed model fits quite nicely with our evolving interpretations, and we have now included it in the Discussion (paragraph seven). We have now also included a brief mention of the disproportionately slow inactivation kinetics for X31 HAs on free virions in the revised legend for Figure 4.

*8) The simulation model uses the parameter* k*, the rate of activation of independent HA. In the present manuscript, a somewhat different rate is used compared to the 2013* eLife *work where* k *= 0.0025 /s for all simulations of X31 (H3N2) influenza. Here* k*_sim_ was changed so that it matched experiments in Otterstrom 2014 (paragraph two, subheading “Fab inhibition of H3 HA” and paragraph two, subheading “Fab inhibition of H1 HA”). Question: is the resulting fit rather insensitive to changes of 5-10 in this rate? This point may need some discussion.*

This is correct; the resulting fit is completely insensitive to the changes in this rate. We have added a statement to this effect in the Computer simulations section of the Methods, within the subheading titled *k*_sim_. We have also added a similar brief statement in the legend to Figure 3, and a more extensive explanation in the Methods for requiring different *k*_sim_ values to recapitulate Otterstrom et al. data in the light of our new conclusions (*k*_sim_ subheading in the Methods). The bottom line is that hemifusion delays measured by Otterstrom et al. (2014) are comparable to our own measurements with X31HA-Udorn virions (Ivanovic et al., 2013). The value for *k*_sim_ we had used previously was quite arbitrary (as we had stated at the time) because we had no information about the frequency of unproductive events. It is the new determination of this relative frequency that allowed us to relate the *k*_sim_ value to the rate of individual HA insertion events.

9) Abstract: comparing only 2 strains and one mutant might be too few to make a general statement about HA evolvability. Perhaps this statement should be softened.

Yes, thank you. We have modified this statement in the Abstract.

*Specific suggestions on the figures:Figure 1 should be revised to represent each of the parameters of the modeling more clearly. A schematic of the model that explicitly shows how each of the main parameters (N_h_,* k*_sim_,* f*_np_, antibody Fab, and patch size) plays a role in the process of fusion is important. Figure 1 shows prefusion productive vs. non-productive HA refolding, but a more comprehensive schematic overview of all of the key steps that are being modeled would be useful in order to quickly grasp what is being simulated.*

We have generated a new Figure 2, which defines all the modeled parameters using simple 2D diagrams of target-membrane contact areas. Figure 2 legend defines symbols we use to represent active HAs, inactive HAs, unproductive HAs, inserted HAs, inactivated HAs, etc. Except in the panel B, where we show hemifusion delays in seconds to illustrate effects of *k*_sim_, we show hemifusion delays as ratios (to illustrate *k*_sim_-independent kinetic effects of changing the modeled parameters). Panel A defines the patch size, panel B – *k*_sim_, panel C – N_h_ = 4 to 6 (N_h_ = 3 is illustrated in the remainder of the panels), panel D – fraction inactive and unproductive HAs and how they combine in the common parameter *f*_np_.

Leave out the HA structural details in an overall schematic of the modeling, although it is still useful for showing productive vs. non-productive HA.

See the new Figure 2 and the preceding comment.

Use a simple representation for active HA trimers, inactive HA trimers, or Fab-inhibited trimers.

See the new Figure 2 and the preceding comment.

Make the schematic multiple panels, displaying the HA molecules (active and inactive) at the interface of the viral and cellular membranes.

See the new Figure 2 and the preceding comment.

*Within each panel one can highlight/define two model parameters – the fraction of non-participating trimers (*f*_np_) and the patch size (PS).*

See the new Figure 2 and the preceding comment.

*Use multiple panels to show individual HA trimers being activated (to highlight/represent* k*_sim_) and also use the number of panels to indicate the number of activated HA trimers required for driving membrane fusion (e.g three activation panels for N_h_ of 3).*

See the new Figure 2 and the preceding comment.

*There could be an indicator for the overall fusion delay time above the panels that helps tie this observable to the discrete steps affected by* k*_sim_,* f*_np_ and N_h_.*

See the new Figure 2 and the preceding comment.

*Split Figure 1 into two figures. The first could show the hemifusion and hemifusion delay predictions and the second could show the* N*_gamma_ and* k*_gamma_. There may be other ways to divide these up (e.g. separated by patch size).*

We assume that this comment refers to the original Figure 2, current Figure 3. We have divided the figure by moving PS55 panels to supplementary information. We feel that it is helpful to have the four plots (hemifusion yield, hemifusion delay, *N*_gamma_ and *k*_gamma_) aligned with each other in the same figure to facilitate their direct comparisons. We have also added illustrations of what is being simulated using patch-size diagrams introduced in the new Figure 2.

In Figure 5, consider putting the PS55 panels into supplementary information. The PS121 are really the most interesting with comparisons to the experimental data.

Current Figure 6 – we have moved the PS55 panels into supplementary information as suggested. We have also added illustrations of what is being simulated using patch-size diagrams introduced in the new Figure 2.

*Figure 4 and Figure 6: although the figure has boxes around the parameter sets of N_h_ and* f*_np_ that agree best with the Fab inhibition data, the main point is still mostly buried. The figures could be simplified somehow, by de-emphasizing the majority of the bars that are not relevant to the main point being made (e.g. using different colors or some other approach).*

Current Figure 5 and Figure 7 – we have changed the coloring scheme as suggested and moved PS55 panels into supplementary information. We have also added illustrations of what is being simulated using patch-size diagrams introduced in the new Figure 2.

Figure 7: Same as Figure 5. Consider moving the left panels to supplementary figures and just show the predictions with the experimental data. The authors should discuss why they think the experimentally observed hemifusion delay could increase as a function of the bound Fabs in contrast to the model.

Current Figure 8 – we have moved the PS55 panels into supplementary information as suggested. We have also added illustrations of what is being simulated using patch-size diagrams introduced in the new Figure 2.

We find that the apparent increases in hemifusion delay for H1N1 experiments are rather unconvincing for the particular data set in question given all the noise in the data, so we would rather not speculate on potential causes for something we are not convinced is real.

*Figure 8: panel A could be removed if a comprehensive schematic is shown in Figure 1. In this current panel 8A the model parameters are insufficiently represented.* f*_np_ is shown as an arrow, similar to k_fp_ (or* k*_sim_). The participating HA trimers go immediately to two postfusion trimers, indicating N_h_. However, this is too similar to a standard representation of fusion that loses the key point that there are multiple independent HA activation events contributing to the accumulation of N_h_ trimers, which can then drive fusion. Panel 8B also could use some revision to make the H3N2 and H1N1 comparison more clearly. Question: what do the filled diamonds represent?*

Current Figure 9 – we have removed the original Figure 8 diagram per your suggestion. We have generated an entirely new model Figure 9. The main mechanistic conclusions are now summarized in panel A using patch-size diagrams (as defined in Figure 2) to illustrate both N_h_ and *f*_un_. We use vertical bars to illustrate the theoretical fusion yields. We show mean first-event delays (in seconds) to illustrate the effects of k_e_, which we contrast with the overall hemifusion-delay values just below them. Panel B now illustrates consequences of the evolved fusion mechanistic features for the two analyzed strains on their susceptibilities to neutralization. (Answer: filled diamonds are no longer used in the current version of the figure).